# Low biodegradability of particulate organic carbon mobilized from thaw slumps on the Peel Plateau, NT, and possible chemosynthesis and sorption effects

Sarah Shakil[1], Suzanne E. Tank[1], Jorien E. Vonk[2], Scott Zolkos[1,3]

[1]Department of Biological Sciences, University of Alberta, Edmonton, Canada
[2]Department of Earth Sciences, Vrije Universiteit, Amsterdam, The Netherlands
[3]current address: John A. Paulson School of Engineering and Applied Sciences, Harvard University, Cambridge, MA, 02138, USA

*Correspondence to*: Sarah Shakil (shakil@ualberta.ca)

**Abstract.** Warming and wetting in the western Canadian Arctic are accelerating thaw-driven mass wasting by permafrost thaw slumps, increasing total organic carbon (TOC) delivery to headwater streams by orders of magnitude primarily due to increases in particulate organic carbon (POC). Upon thaw, permafrost carbon entering and transported within streams may be mineralized to $CO_2$ or re-sequestered into sediments. The balance between these processes is an important uncertainty in the permafrost-carbon-climate feedback. Using aerobic incubations of TOC from streams affected by thaw slumps we find that slump-derived organic carbon undergoes minimal (~ 4%) oxidation over a 1-month period, indicating that this material may be predominantly destined for sediment deposition. Simultaneous measurements of POC and dissolved organic carbon (DOC) suggest that mineralization of DOC accounted for most of the TOC loss. Our results indicate that mobilization of mineral-rich tills in this region may protect carbon from mineralization via adsorption to minerals and promote inorganic carbon sequestration via chemolithoautotrophic processes. With intensification of hillslope mass wasting across the northern permafrost zone, region-specific assessments of permafrost carbon fates and inquiries beyond organic carbon decomposition are needed to constrain drivers of carbon cycling and climate feedbacks within stream networks affected by permafrost thaw.

## 1 Introduction

Permafrost soils comprise the single largest pool of terrestrial organic carbon (OC) (Schuur et al., 2015; Hugelius et al., 2014), half of which may be vulnerable to rapid mobilization into modern biogeochemical cycles via abrupt thaw processes (Turetsky et al., 2020; Olefeldt et al., 2016). Permafrost dissolved organic carbon (DOC), typically defined as compounds < 0.7µm, is often highly susceptible to biotic mineralization into $CO_2$ within aquatic systems (Vonk et al., 2015a; Littlefair and Tank, 2018; Abbott et al., 2014; Mann et al., 2015). Abrupt thaw can mobilize orders of magnitude more particulate organic carbon (POC, typically > 0.7µm) than DOC, yet the biodegradability of permafrost POC is not well understood (Shakil et al., 2020; Tank et al., 2020; Vonk et al., 2015b).

Suspended particles can be important sites for mineralization (Attermeyer et al., 2018) or mineral protection (Hemingway et al., 2019; Opfergelt, 2020; Groeneveld et al., 2020). In addition to molecular composition and a host of environmental factors that typically affect organic matter decomposition (e.g., microbial activity, nutrient availability) (Kothawala et al., 2021), mineralization of POC in stream networks depends on transport vs. deposition. When settled out, mineralization of POC can

be reduced by ~50% (Richardson et al., 2013) or more, particularly if contained in anoxic sediments (Peter et al., 2016), though carbon release can shift to be in the form of methane (Schädel et al., 2016). Fractions of POC with different density and size therefore not only experience a different settling and transport trajectory, but also may have differing processes and rates affecting OC dynamics (Tesi et al., 2016). If biodegradability varies across size and density fractions, this could alter realized mineralization during transport relative to measurements on bulk OC (Tesi et al., 2016).

Warming and intensifying precipitation across the ice-rich terrain of the Peel Plateau in western Canada has triggered an acceleration of thaw-driven landscape erosion in the form of retrogressive thaw slumps (hereafter, slumps) (Kokelj et al., 2021). Thaw slumping along stream sites in this region can increase total organic carbon (TOC) yields by orders of magnitude, almost entirely due to increases in POC (Shakil et al. 2020). Slump-derived DOC in the region is relatively more labile than background DOC, as shown by chemical composition and incubations (Littlefair et al., 2017; Littlefair and Tank, 2018).

Slump-derived POC chemical composition suggests lower bioavailability as POC sources shift from the active layer and some periphyton material to Pleistocene-aged organic carbon and petrogenic organic carbon mobilized from permafrost (Shakil et al., 2020; Bröder et al., 2021). However, POC bioavailability has not been experimentally assessed. Given that slump-derived carbon occurs almost entirely as POC in this region (Shakil et al., 2020), understanding the fate of this carbon remains a critical knowledge gap.

Our objectives were to assess the potential for slump POC to be mineralized to $CO_2$ during transport in streams. To do this, we undertook experimental incubations to: (a) determine whether slump-POC differs in biodegradability from POC present in unimpacted waters; and (b) quantify and assess the biodegradability of slump-POC fractions relative to their transport potential. This work provides insight into the fate of an understudied component of permafrost-mobilized OC.

## 2 Methods

### 2.1 Region and field sampling

Slumping occurs across the Peel Plateau (Fig. 1) and typically mobilizes terrestrial material from three distinct sources: (1) Pleistocene-age tills that have remained preserved within permafrost since deposition by the Laurentide Ice Sheet and subsequent permafrost aggradation; (2) Holocene-age permafrost developed from tills following active layer deepening and/or slumping in previous warm periods, followed by permafrost aggradation during a cooler climate; and (3) a contemporary active

layer. Thus, the relative contribution of biogeochemical substrate from these three terrestrial sources to streams can depend on thaw depth (Shakil et al., 2020; Bröder et al., 2021). Source composition can also vary west-east as vegetation (elevation) and geology change along this gradient (Duk-Rodkin and Hughes, 1992; Norris, 1985).

Sampling occurred during July-August, within the Stony Creek and Vittrekwa River watersheds of the Gwich'in Settlement Region on the Peel Plateau (Fig. 1). In 2015, substrate from streams near and within three slump sites (HA, HB, HD) was used to test if mobilization of slump-POC and nutrients affect biodegradability of OC in streams. Stream water samples were obtained from: channelized runoff within each thaw slump (IN); a downstream location where all runoff entered the valley-bottom stream (DN); and an unimpacted reference stream upstream of slump inflow (UP) (Fig. 1b). Site HD-UP experienced some encroachment of slump runoff, and thus was not a fully unimpacted site (Fig. S5). In 2016, substrate was collected upstream, within, immediately downstream, and 2.79 km downstream of slump SE to assess variations in biodegradability with transport potential. In 2019, substrate was collected within and downstream of slump FM3 to follow-up results from 2015 and 2016 (details below). Slump sites had varied elevation and morphology, with maximum headwall heights (Fig. 1a) ranging from 7.1 to 23m (see Shakil et al., 2020). All samples were processed (i.e., filtered) within 24 hours of collection, apart from within-slump and downstream samples used for adding particles to "unfiltered" treatments in 2016 that were stored in the dark at 4°C until the start of the experiment. Experiments were initiated within 24 hours (2015, 2019) or 48 hours (2016) after processing. The extra hold time for the 2016 experiment was due to the extra time needed for size fractionation of samples (see below and supplementary S2). For further sample collection details, see supplementary S1.

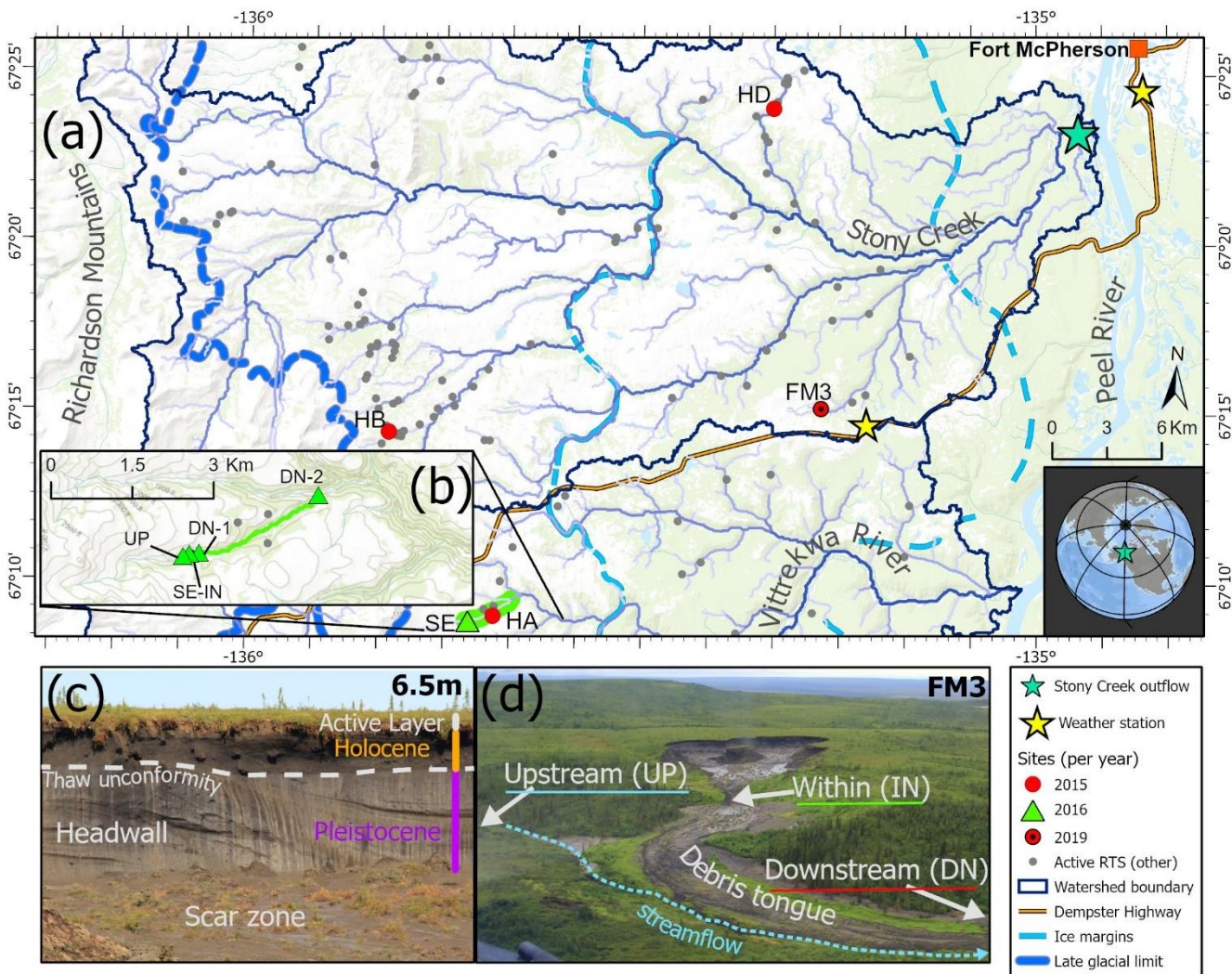

**Figure 1.** (a) Location of slump sites sampled for experiments. (b) 2016 sampling points (green triangles) show sampling locations along a transect which began just upstream of the point where slump SE runoff enters a valley bottom stream. Panel (c) shows headwall units of a slump. Panel (d) shows stream sampling points relative to a slump site, as sampling design used in the 2015 experiment. HA, HB, HD, SE and FM3 are slump site IDs. Active RTS features are from Segal et al. (2016). Former glacial limits of the Laurentide Ice Sheet (ice margins and late-glacial limit) are delineated from Duk-Rodkin and Hughes (1992). Service Layer credits: (1) World Topographic Map: Northwest Territories, ESRI, © OpenStreetMap contributors 2020, HERE, Garmin, USGS, NGA, EPA, USDA, NPS, AAFC, NRCan. Distributed under the Open Data Commons Open Database License (ODbL) v 1.0. (2) World Ocean Base: Esri, Garmin, GEBCO, NOAA NGDC, and other contributors. (3) World Continents.

**2.2 Biodegradation experiments**

**2.2.1 Effects of POC source, dissolved constituents, and settling (2015)**

To test the effect of POC source (Table 1), we incubated unfiltered upstream water (upstream POC; treatment "UU") and filtered upstream water with a 2 mL addition of slump runoff (slump POC; "SU") in 120 mL glass serum bottles for 7 days at ca. 20°C in the dark, with continuous end-over-end rotation (4 rpm; Richardson et al. 2013) (supplementary S2, Fig. 2). Control bottles accounted for DOC contained in filtered upstream water alone (no POC control; "UF"). Additionally, we tested for: (a) the effect of particle deposition by allowing a replicate set of SU bottles to settle out ("SS"); and (b) the slump-derived release

of additional solutes (e.g., nutrients) by mixing slump POC with filtered downstream water ("SD"). Bottles were filled to have no headspace.

**2.2.2 Variability as a function of transport potential (2016)**

SE within-slump runoff was split into three sieve size fractions (63 – 2000 µm, 20 – 63 µm, < 20 µm) by sieving a 0.5 mL aliquot and adding the resultant size fractions to filtered downstream water in 60 mL glass BOD bottles (supplementary S2,

Fig. 2). An unfractionated control (0.5 mL in 60 mL downstream water) was also created, and bottles were incubated for 8 days in the dark at ca. 20°C as above. Since relative concentrations of each size fraction were maintained, the <20 µm fraction had orders of magnitude greater total suspended solid (TSS) concentrations than the larger two fractions (Table S2). We also incubated filtered and unfiltered (but diluted, Table S2) stream water from sites upstream, downstream, and 2.79 km downstream of SE to accompany size fraction incubations. We characterized POC differences between size fractions using:

(a) $^{14}$C age, (b) percent POC (%POC; POC : TSS), and (c) absorbance and fluorescence spectra of base-extracted particulate organic matter (BEPOM) (Osburn et al., 2012)(supplementary S3). Bottles were filled to have no headspace.

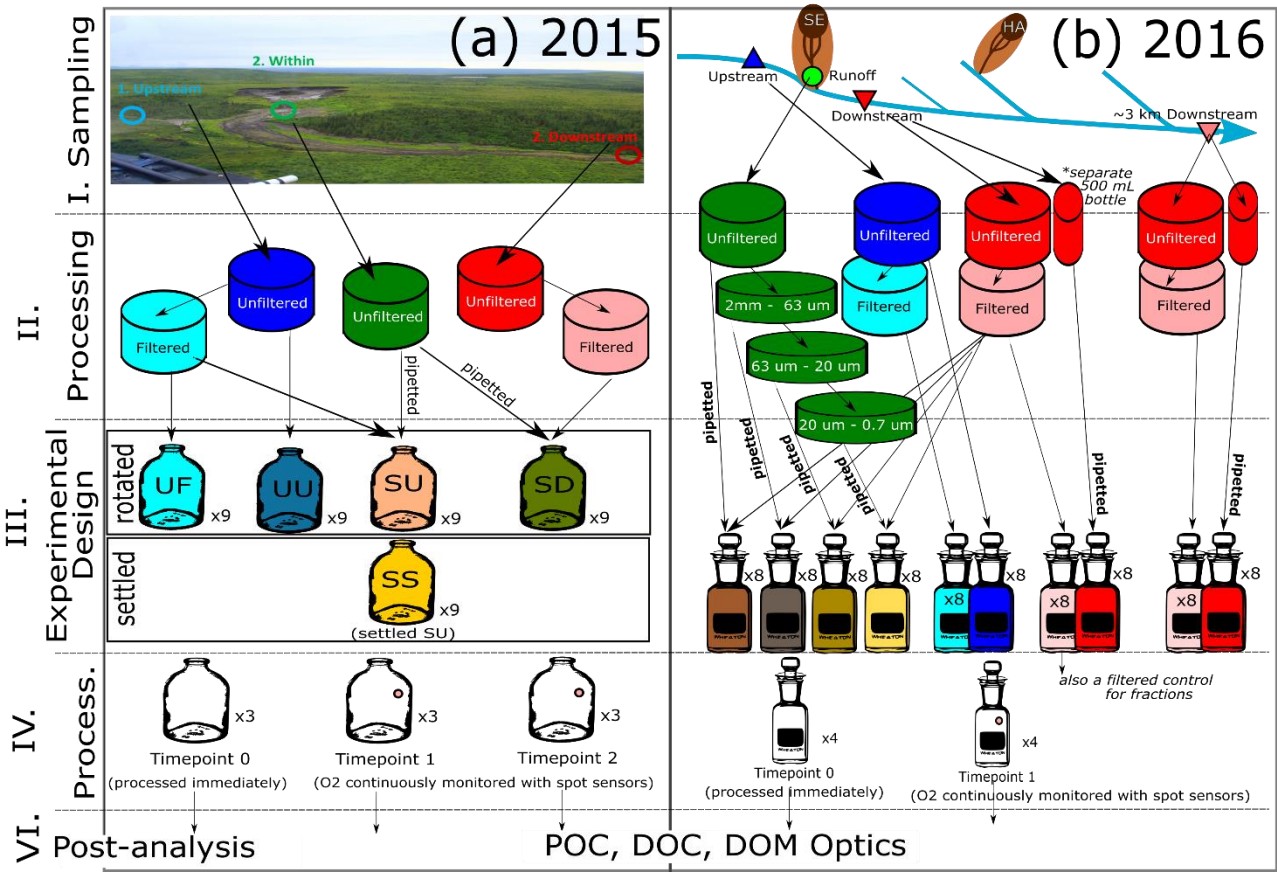

**Figure 2.** Flow chart for processing of (a) 2015 and (b) 2016 experiment. Note, timepoint two for 2015 occurred shortly after timepoint one due to rapid oxygen loss. Due to this, and the fact that some bottle replicates had to be removed because of anoxia, only data for timepoint one are presented in the main manuscript. Analyses show particulate (POC) and dissolved (DOC) organic carbon and dissolved organic matter (DOM) optics (SUVA$_{254}$). Flow chart for 2019 is provided in Appendix D.

### 2.2.3 Measurements

We measured: (a) concentrations of DOC, POC, and TOC, and SUVA$_{254}$ (an optical proxy for dissolved organic matter (DOM) aromaticity) (Weishaar et al., 2003) at the beginning and end of incubations; and (b) O$_2$ concentrations approximately daily (PreSens, Fibox4, SP-PSt3-NAU-D5-YOP) to provide insight on rate of change (Richardson et al. 2013). We initially assumed heterotrophic breakdown of OC would be the dominant O$_2$ consumption pathway, as respiratory quotients across several freshwater sites have been noted to vary around 1 (Berggren et al., 2012). Incubation O$_2$ concentrations presented never dropped below 2 mg L$^{-1}$; a threshold well above O$_2$ limiting concentrations for different bacterial species (Stolper et al., 2010 and references therein). One exception was one of four replicates for the SE unfractionated treatment, which was removed and

replaced with the mean of replicates for statistics. Our experiments aimed to mimic conditions downstream of slump inflows, thus slump-affected incubations had POC concentrations ranging from 1.4 to 18.6 times DOC concentrations. Samples for TSS concentration were collected alongside POC. Further details are available in supplementary S3.

## 125 **2.2.4 Follow-up experiments**

To assess processes that could consume $O_2$ and/or generate OC (due to $O_2$ losses coupled with OC gains observed in 2015 and 2016) we undertook two follow-up experiments. First, we combined 0.15 mg of sterilized HD debris tongue sediments (collected in 2016)(Zolkos and Tank, 2020) with 18.2 Ω Milli-Q (MQ) water to assess abiotic $O_2$ loss. MQ water was sourced from a machine with a carbon filter and was quality controlled to have less than 10 ppb TOC. We incubated 60 mL glass
biological oxygen demand (BOD) bottles on a shaker table in the dark at ca. 20°C for 7 days, monitoring $O_2$ as above. Second, we measured change in dissolved and particulate inorganic carbon (DIC, PIC), in addition to change in DOC and POC, in an incubation combining FM3 slump runoff with downstream water, including sterilized replicates. The treatments were designed to replicate treatment "SD" in 2015. Sterilization was achieved by autoclaving and adding $ZnCl_2$ as a poison and was validated using plate counts (supplementary S4). We hypothesized that chemosynthesis associated with nitrification and sulfide
oxidation (eqns. 1-2) could generate OC and so we additionally measured dissolved inorganic nitrogen ($NH_4^+$, $NO_3^-$, $NO_2^-$) via automated colorimetry and sulfate ($SO_4^{2-}$) via ion chromatography at the Canadian Association of Laboratory Accreditation (CALA)-certified Biogeochemical Analytic Service Laboratory (BASL; University of Alberta, further details in supplementary S3.3).

$$NH_3 + 1.5O_2 \leftrightarrow NO_2^- + H^+ + H_2O \quad \text{(1a, Stumm and Morgan, 2012)}$$
$$NO_2^- + 0.5O_2 \leftrightarrow NO_3^- \quad \text{(1b, Stumm and Morgan, 2012)}$$

$$CO_2 + H_2S + O_2 + H_2O \leftrightarrow [CH_2O] + H_2SO_4 \quad \text{(2, Klatt and Polerecky, 2015)}$$

Note that equation 2 is a general equation of chemolithoautotrophic reduced sulfur oxidation that can have a variable stoichiometry and assumes sulfur oxidizing bacteria exclusively produce $SO_4^{2-}$, rather than both $SO_4^{2-}$ and $S^0$ (Klatt and Polerecky, 2015; Nelson et al., 1986). Equation 3 shows a net reaction for a model of pyrite oxidation at circumneutral-pH (Percak-Dennett et al., 2017). We note that pH in the streams in this study can be quite variable but tend to be circumneutral, often varying around pH 7 and mostly ranging from pH 6 − 8 (see supplementary data of Shakil et al. 2020). This sulfide
oxidation can generate sulfuric acid that can weather carbonates (e.g., eq. 4) or silicates (Zolkos and Tank, 2020; Zolkos et al., 2020).

$$FeS_2 + 3.75O_2 + 3.5H_2O \leftrightarrow \ + 2H_2SO_4 + Fe(OH)_3 \quad \text{(3, Percak-Dennett et al., 2017)}$$
$$H_2SO_4 + 2(Ca, Mg)CO_3 \leftrightarrow 2(Ca^{2+}, Mg^{2+}) + SO_4^{2-} + 2HCO_3^- \quad \text{(4, Calmels et al., 2007; Zolkos and Tank, 2020)}$$

Sediments in sterilized and unsterilized bottles were additionally characterized using X-ray diffraction (XRD) (supplementary S3.4), while absorbance and fluorescence spectra of BEPOM and DOM were assessed at the beginning and end of the experiment (see supplementary for further details).

## 2.3 Data analyses

Two-way ANOVAs, with site and treatment as fixed effects, were used to assess the effect of POC presence (UF vs SU), source (UU vs SU), dissolved matrix (SU vs SD), and settling (SU vs SS) on percent changes in OC (DOC, POC, and TOC), DOM aromaticity (SUVA$_{254}$) and O$_2$ loss rate. One-way ANOVAs were also used to assess differences in the aforementioned changes between size fractions (2016 experiment). Two-way ANOVAs were used to assess the effect of distance and filtration (fixed effects) for the 2016 transect experiments. Significant ANOVA tests were followed up with Tukey-adjusted pair-wise

t-tests (Zar, 2010). We also calculated 95% confidence intervals to evaluate whether OC changes significantly differed from zero.  Principal components analyses were used to visualize differences in optical indices between size fractions (2016), and changes in DOM and BEPOM (2019), following calculation of SUVA$_{254}$ (DOM only; Weishaar et al., 2003; Poulin et al., 2014) and slope ratios (Helms et al., 2008) on absorbance data, and humification indices (HIX; Ohno, 2002), biological indices (BIX; Huguet et al., 2009), the proportion of peaks C and M (Coble, 2007), and previously identified BEPOM peaks (Shakil

et al., 2020) on fluorescence data (supplementary 3.6 – 3.9). To assess factors controlling in situ CO$_2$ and O$_2$ dynamics we calculated departures of O$_2$ and CO$_2$ from atmospheric equilibrium (Vachon et al. 2019) using 2015 in situ measurements of dissolved O$_2$ at several slump sites (Shakil et al., 2020) and coupled CO$_2$ departures (Zolkos et al., 2019). For further details, see Appendix A.

## 3 Results

### 3.1 Effects of POC presence and source (2015)

Across experiments, declines in POC were not observed, and in some cases, POC increased (Figure 3). Slump runoff addition into filtered upstream water (SU) did not significantly alter %ΔDOC (the percent change in DOC from beginning to end of the experiment), ΔSUVA$_{254}$ (absolute change in DOM aromaticity), or %ΔTOC (p>0.05; Table A1, Figure 3) relative to the upstream filtered control (UF). Similarly, POC source (slump [SU] vs. unfiltered upstream [UU]) did not significantly affect

%ΔTOC or ΔSUVA$_{254}$ (p>0.05, Table A1, Figure 3).  However, %ΔPOC was significantly lower when particles were sourced from slump runoff (SU vs. UU; p<0.001, Table A1), potentially because particle concentrations were orders of magnitude lower in upstream bottles (Table S2). DOC increased in the presence of upstream particles (UU) but decreased in the presence of slump particles (SU), though this difference was marginally insignificant (Table A1, p = 0.053). Despite no effect on %ΔTOC (Table A1), the addition of slump particles (SU) did significantly increase rates of O$_2$ consumption compared to

185 upstream filtered and unfiltered treatments (UF and UU; Figure 3 a-c), though this effect was dependent on slump site (significant interactions, Table A1; note lack of increase for site HD where slump runoff encroached the upstream site).

**Table 1.** Summary of experiments and main results with reference to figures and text sections for details.

| Year | Field Sampling | Exp. Details | #days | Test | Treatments | Main result | Considerations |
|---|---|---|---|---|---|---|---|
| 2015 | UP, IN, and DN of slumps HA, HB, HD | Sec. 2.2.1, Figure 2 | 7 | POC presence | **SU** *(slump in filtered upstream) vs.* **UF** *(filtered upstream)* | (1) between treatments: no effect on %ΔTOC but increased $O_2$ loss; (2) within treatments: no sig. TOC loss *(Fig 3, Table A1)* | |
| | | | | POC source | **SU** vs. **UU** *(unfiltered upstream)* | | |
| | | | | Dissolved constituents | **SU** vs. **SD** *(slump in filtered downstream)* | (1) Between treatments: no effect on %ΔTOC or $O_2$ loss; (2) within treatments: no sig. TOC loss *(Fig 3, Table A1)* | |
| | | | | Settling | **SU** vs. **SS** *(SS off rotator to allow settling)* | (1) Between treatments: no effect on %ΔTOC, reduced $O_2$ loss; (2) within treatments: no sig. TOC loss *(Fig 3, Table A1)* | |
| 2016 | UP, IN, DN, and 2.79 km DN of slump SE | Sec. 2.2.2, Figure 2 | 8 | Biodegradability vs. transport potential | Sieve size fractions: (1) 2000-63µm (2) 63-20µm (3) <20µm). + unfractionated reference | No sig. diff. in %Δ TOC or %Δ POC changes between size fractions but sig. TOC gain and largest DOC loss for particles <20 µm *(Fig 3, Table A2)* | ↓ [TSS] in two larger size fractions relative to smallest, two largest also within MQ error |
| | | | | Transect validation | unfiltered and filtered DN vs. 2.79 km DN. + UP reference | Change in downstream distance: no effect on %ΔTOC; within treatments: no sig. TOC loss *(Fig 3, Table A3)* | UP and 2.79 km DN within MQ blank error |
| 2018 | HD debris tongue material | Sec. 2.2.4 | 7 | Abiotic $O_2$ loss | Sterilized debris tongue sediments in Milli-Q (MQ) water vs. MQ control | Rapid $O_2$ loss in absence of microbial activity *(Fig S3a)* | MQ may accelerate weathering, HD debris sediments an extreme weathering endmember |
| 2019 | IN and DN of slump SC | Sec. 2.2.4, Fig. D1 | 27 | paired inorganic carbon changes & chemo-lithotrophy | Unsterilized mimic of **SD** (2015 treatment) vs. sterilized **SD.** + MQ blanks | Prior gains not replicated but only ~4% of TOC mineralized, greater sulfate gains and nitrification in | SC slump in a different landscape type than SE and HA where prior gains were observed, |

| unsterilized treatments *(Fig 5)* | sterilization treatments may not be a true abiotic control |


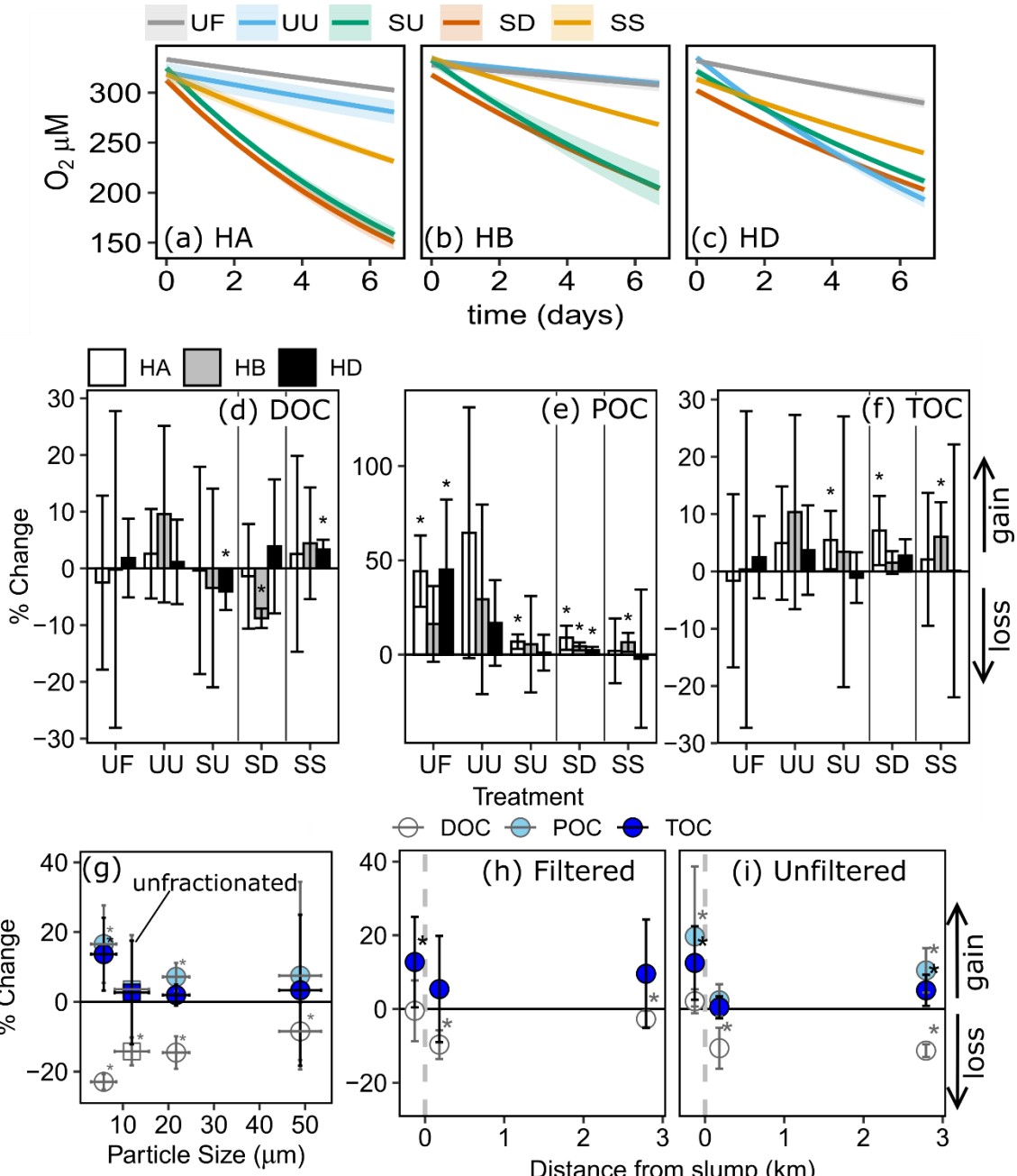

**Figure 3.** (a-c) Modelled (line) $O_2$ (mg $L^{-1}$) across combinations of source material and settling effects. Percent change in DOC, POC, and TOC in comparison to: (d-f) combinations of source material and settling effects; (g) geometric mean particle size; and (h-i) distance from slump site. Vertical errors are 95% confidence intervals with asterisks marking significant differences from zero. Horizontal errors (e) are particle size geometric standard deviation. Codes (a-f): filtered (UF) and unfiltered (UU) upstream, slump material in upstream (SU) and downstream (SD) filtrate, and SU settled out (SS). HA, HB, and HD are slump sites. For measured $O_2$ data, see Fig. S1 (2015), Fig. S2 (2016), Fig. S3 (2018-19).

### 3.2 Effects of background dissolved constituents and settling (2015)

Changing filtrate to downstream water, which has higher ion and nutrient concentrations (Shakil et al. 2020), had no significant effect on any parameter measured in the experiment (SU vs. SD, Table A1). Allowing slump particles to settle (SU vs SS) did not affect %ΔTOC but did significantly reduce $O_2$ consumption rates. Mean DOC concentrations also switched from decreasing (SU) to increasing (SS) (Figure 3d), though the difference was not significant (p=0.10, Table A1).

### 3.3 Variability dependent on transport potential (2016)

Based on [14]C age, %POC, and the relative contribution of fluorescent peak C (Coble, 2007), larger particle size fractions appeared to be associated with fresher terrestrial-origin organic matter than smaller size fractions (Appendix B). POC associated with particles < 20 µm dated to > 27,000 cal BP, while POC associated with particles ranging from 20 – 63 µm and 63 – 2,000 µm dated to ~ 19,600 cal BP and ~8,000 cal BP respectively (Table B1). The majority of POC (73%, Table B2) was associated with particles less than 20 µm.

Although bottles containing particles <20 µm displayed significant gains in TOC (95% CI; Figure 3g-i) ANOVA analyses did not uncover a significant difference in %ΔTOC or %ΔPOC between size fractions (Table A2). DOC losses occurred in all treatments downstream of slumps, to a greater degree when particles were present (i.e., unfiltered treatments) (Figure 3h-i, p<0.05 Table A3), and were significantly greater when particles were <20 µm (p<0.05, Figure 3g, Table A2). Increases in SUVA$_{254}$ were also significantly greater for <20µm particle treatments (p<0.05 Table A2 and C1), as were TSS concentrations

(Table S2). %DOC loss was also significantly greater 2.79 km downstream of slump SE compared to immediately downstream (Table A3).

### 3.4 $O_2$ vs. carbon

Change in $O_2$ and TOC generally did not follow the 1:1 trend expected if heterotrophic respiration dominated metabolic processing (Figure 4 a-c). The greatest deviations from 1:1 were observed in treatments containing slump runoff, where despite

large losses in $O_2$, we saw non-significant changes to gains in TOC. Increases in TOC from upstream, filtered, and 2.79 km downstream bottles were within the range of experimental blanks (Figure 4c).

Although the rate of $O_2$ consumption within and across experiments was generally greater in treatments with greater initial TSS (Figure 4d), there was no consistent relationship between TOC changes and initial TSS (Fig. 4e). However, some of the greatest TOC increases occurred during incubations of slump SE particles <20 µm and slump HA particles in upstream (SU)

and downstream (SD) filtrate, treatments amongst those withthe greatest initial TSS (Figure 4a, b, e).

In-situ comparisons of $O_2$ vs. $CO_2$ showed within-slump samples to have the greatest excess $CO_2$, with several samples substantially departing from the 1 $CO_2$: -1 $O_2$ stoichiometry associated with heterotrophic respiration (Figure 4f). In contrast, several downstream and upstream sites displayed measurements close to atmospheric equilibrium for $CO_2$ but were supersaturated for $O_2$, potentially due to temperature changes and lower solubility of $O_2$ (Vachon et al., 2020).


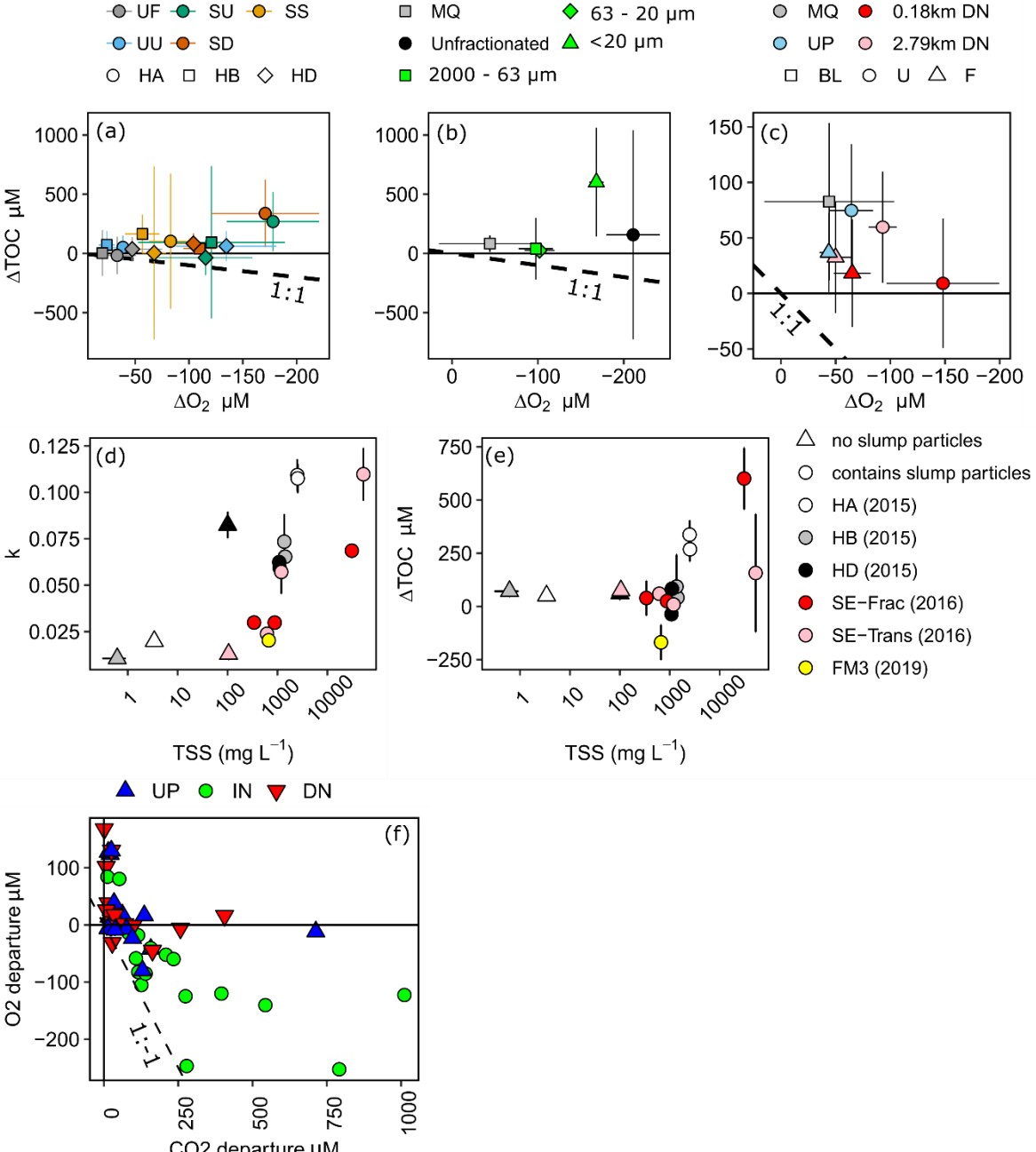

**Figure 4.** Concentration changes in TOC vs. changes in dissolved $O_2$ for: (a) 2015 experiments; (b) 2016 site SE fractionation experiment; and (c) 2016 site SE transect experiment. Dashed lines in a-c represent predicted loss of OC for respiratory quotient=1 and error bars show 95% confidence intervals. (d-e) Exponential rate of $O_2$ consumption (k) or changes in TOC vs. initial TSS for treatments across experiments, 235 excluding filtered, settled, and sterilized treatments from a-c. SE-Frac indicates the fractionated treatments from panel (b). (f) Departures of $O_2$ and $CO_2$ from atmospheric equilibrium in samples collected upstream (UP), downstream (DN) and within (IN) a series of slump sites on

the Peel Plateau. Error bars in (d) and (e) show standard error of the mean. Codes in (a): filtered (UF) and unfiltered (UU) upstream, slump material in upstream (SU) and downstream (SD) filtrate, and SU settled out (SS). HA, HB, and HD are slump sites.

## 3.5 Follow-up experiments

### 3.5.1 Sterilized debris sediments in Milli-Q water (2018)

Oxygen was completely consumed (~226 µM) in bottles containing sterilized HD debris tongue material suspended in MQ water within 4-5 days, exceeding the $O_2$ loss rates previously observed (Table S1, Fig. S3a). Bottles containing sterilized sediments had lower pH (5.52 – 6.09) following incubation than MQ controls (6.53 – 6.91).

### 3.5.2 Inorganic carbon changes and potential chemolithoautotrophy (2019)

Oxygen consumption occurred in sterilized treatments (-15 ± 6 µM, mean ± 95% CI, t = 27 days) but was more pronounced in unsterilized bottles (-124 ± 15 µM). The pronounced $O_2$ decline in unsterilized bottles was accompanied by a significant loss of DOC (-83 ± 26 µM) and a non-significant loss of POC (-85 ± 261 µM), balancing to a non-significant loss of TOC (-170 ± 262 µM, Fig. 4). Total inorganic carbon (TIC) increased significantly in unsterilized treatments, driven by increases in DIC. In sterilized bottles, modest $O_2$ losses were accompanied by significant DOC losses (-93 ± 51 µM), significant POC gains (141 ± 33 µM), balancing to a modest non-significant gain in TOC (48 ± 66 µM). TIC in sterilized bottles had a minor significant decrease (-25 ± 20 µM), driven by losses in DIC.

Ammonium ($NH_4^+$) decreased from 8.68 ± 0.47 µM to below detection (0.2 µM) in unsterilized bottles, while $NO_3^- + NO_2^-$ increased by 2.58 ± 2.36 and 1.33 ± 1.67 µM, respectively. In sterilized bottles, $NH_4^+$ increased by 6.74 ± 1.80 µM alongside negligible changes in $NO_3^- + NO_2^-$. Sulfate ($SO_4^{2-}$) generation was greater in unsterilized (90 ± 6 µM) than sterilized (54 ± 23 µM) bottles, but for both treatments $SO_4^{2-}$ increased more than would be expected via pyrite oxidation (Figure 5d, based on oxygen stoichiometry in eq. 3). However, the only sulfur-bearing mineral detected in sediments (XRD; 1-5% detection limit) was pyrite (Table S3).

A biplot of PCA components 1 and 2 did not reveal any shifts in BEPOM or DOM optical characteristics during incubation of unsterilized treatments (Figure 5d-e). However, DOM from sterilized treatments shifted towards lower molecular weight (SR, slope ratio), lower aromatic material ($SUVA_{254}$) of greater biological origin (BIX). PC1 separated DOM in sterilized and unsterilized bottles, suggesting the sterilization processes increased the proportion of simple compounds. Since the sterilization process itself appears to increase the proportion of simple compounds, the results caution against its use as an abiotic baseline.

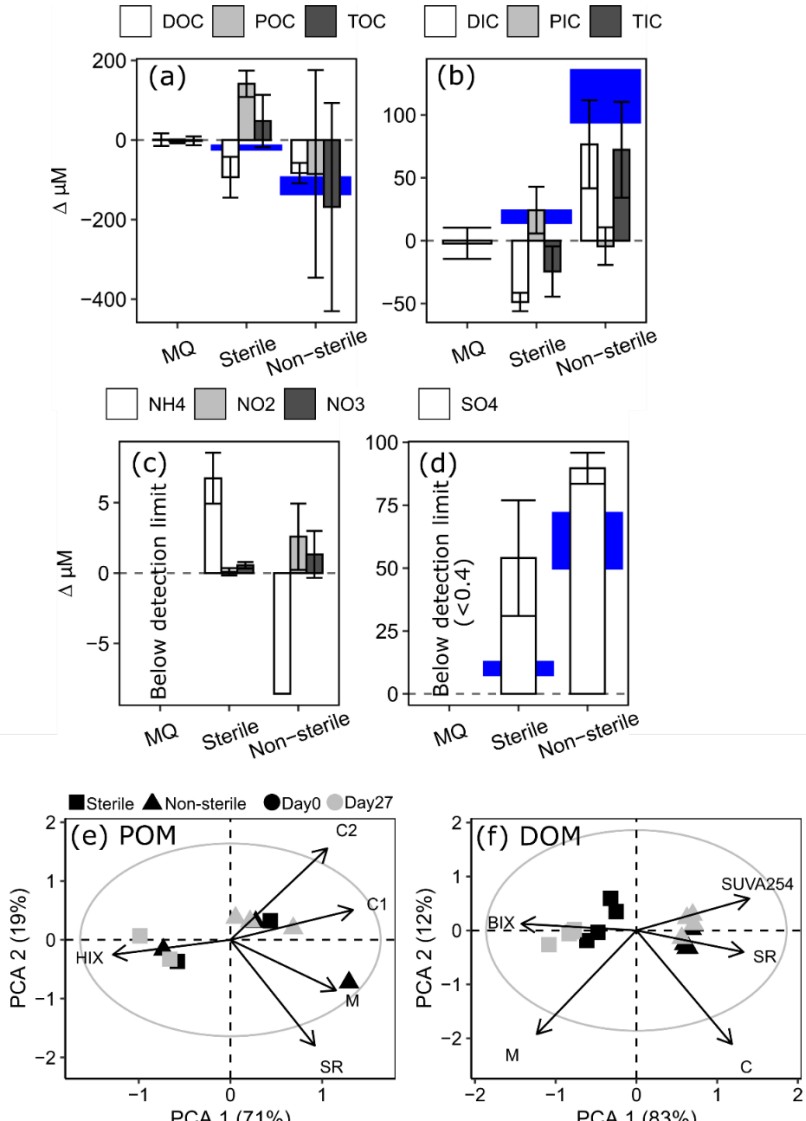

**Figure 5.** Changes in millimolar concentrations of: (a) organic carbon; (b) inorganic carbon; (c) ammonium ($NH_4^+$), nitrite ($NO_2^-$), and nitrate ($NO_3^-$); and (d) sulfate ($SO_4^{2-}$) in 2019 test of interferences. Note difference in scales between panels. Blue shading highlight potential carbon gains or losses based on $O_2$ loss and a respiratory quotient of 1 (a-b) and potential $SO_4^{2-}$ generated from pyrite oxidation (d; eq. 3). Error bars and height of blue shading both show a range representing 95% confidence intervals. (e-f) PCA biplots of components 1 and 2 showing variation in POM (e) and DOM (f) optical properties. Grey circle outlines circle of equilibrium contribution, plot shown in scaling 1. Abbreviations of optical indices are provided in Table S5.

## 4 Discussion

Our incubations, coupled with multiple studies examining slump-POC composition (Shakil et al., 2020; Bröder et al., 2021; Keskitalo et al., 2021) indicate that slump-derived POC in glacial landscapes of western Canada has low biodegradability on

the Peel Plateau. We found no significant losses of POC or TOC or evidence that %ΔTOC increased due to the presence of slump-derived POC. This finding was consistent across slump sites and for varying size fractions and distances downstream of slump inputs. While %ΔTOC did not significantly differ across size fractions of slump SE particles, particles most likely to remain in transport (< 20 µm) enabled the greatest loss in DOC (linked to greater TSS) and significant gains in TOC. A coupled transect experiment showed that downstream of SE, TOC gains were not significant or were within error of blanks (2.79 km downstream), though both downstream treatments had orders of magnitude lower TSS than the <20 µm size fraction treatment sourced directly from slump runoff.

The lack of loss in TOC and POC contrasted with elevated $O_2$ consumption rates in incubations using water collected within and downstream of slumps, except for slump HD, where slump runoff encroached into the upstream site (Fig. S4). Despite a lack of TOC or POC loss, oxygen consumption rates in treatments containing particles were always elevated relative to their DOC controls, highlighting that oxygen consumption could not be solely accounted for by DOC mineralization. Instead, TOC gains suggest the potential for chemoautotrophic carbon sequestration. Further, abiotic processes (e.g., mineral oxidation) appear to have the potential to consume oxygen rapidly enough to decouple oxygen and carbon dynamics from the 1:1 relationship associated with heterotrophic respiration, as suggested by sterile incubations of HD debris material. The excess in situ $CO_2$ concentrations we observe are likely from mineral weathering that can generate $CO_2$ in this system (Zolkos et al., 2018).

Our finding of low POC biodegradability is likely conservative since incubations focus on the most labile period (initial 7-27 days) (Richardson et al., 2013). Our longest incubation (27 d; Figure 5 non-sterile) did not show significant TOC losses, though 95% error spanning losses expected from a 1:1 relationship with $O_2$ suggests that detection of change may be masked by error in POC measurements (Figure 5a). Using ΔTIC from our 2019 experiment as an alternate metric of carbon mineralization, we estimate that a maximum of 4% of the initial TOC pool may have been mineralized within 27 d. The difference in the landscape position and slump morphology of the slump site (SC is further east than slumps HA, HB, and SE) and the longer time period of the incubation (27 days vs. 7-8 days) could have both played a role in the average loss in TOC versus the gain seen in previous comparable experiments where slump material was added to downstream water. Time series experiments on TOC degradation from similarly glacially-conditioned Qikiqtaruk (Herschel Island) indicate that $CO_2$ production tied to organic matter mineralization ceased by the end of a 120-day incubation, while greater than half (~58%) of total $CO_2$ was produced within the first 27 d (Tanski et al. 2019). Assuming a similar rate for OC mineralization, we can scale up our findings beyond the time of our incubation to estimate that ~7% of slump-derived TOC on the Peel Plateau may be mineralized during the entirety of time it is transported in streams. Our findings of minimal TOC mineralization are consistent with measures of little change in $CO_2$ concentrations downstream relative to upstream of slumps (Zolkos et al., 2019), despite orders of magnitude increases in POC and thus TOC (Shakil et al., 2020).

While the upper bound of 7% TOC mineralization calculated from observations in this study is elevated relative to slow rates of mineralization within permafrost (Leewis et al. 2020), it is considerably lower than estimates that greater than 60% of TOC mobilized by hillslope abrupt thaw will be mineralized on a decadal timescale (e.g., Table S1 in Turetsky et al.

2020), which have been based on mineralization rates observed for DOC in Pleistocene Yedoma landscapes (Vonk et al. 2013). Although similarly elevated rates of DOC mineralization have been noted in other studies (e.g. Spencer et al. 2015, Abbott et al. 2014), these DOC-specific findings have not been consistent in landscapes across the Arctic (e.g. Burd et al., 2020; 310   Wickland et al., 2018) likely due to differences in landscape factors and permafrost composition (Tank et al., 2020). Past studies have also generally not included POC within their assessments of permafrost carbon mineralization, even though POC concentrations within thaw streams can be orders of magnitude greater than for DOC (Vonk et al., 2013; Shakil et al., 2020). Notably, percent TOC loss from this study is substantially lower than percent DOC loss previously observed for this study area (Littlefair and Tank, 2018), likely due to substantial differences in biogeochemical processes occurring in the DOC vs. 315   POC pool upon thaw, interaction with mineral surfaces (see further discussion below), and contrasting headwall sources (Shakil et al., 2020). Our loss estimate is comparable to Tanski et al. (2019), who observed 2% to 9% loss rates for incubations of permafrost TOC mixed with seawater and incubated at 16°C for 120 days.  Clearly, our results highlight the need to better understand the relative lability of different organic matter fractions (i.e., DOC vs. POC), and how mineralization rates of these fractions may vary with source (i.e., across landscapes) and receiving environments (lacustrine, fluvial, marine). 320   This increased understanding appears critical for better constraining the magnitude and effective timespan of permafrost carbon degradation in Earth system models.

This study and work by Tanski et al. (2019) both suggest DOC contributes substantially more to heterotrophic $CO_2$ production than POC in glacial margin landscapes even where hillslope thermokarst increases fluvial POC by orders of magnitude (Shakil et al. 2020). This seemingly contrasts protection of DOM by adsorption to mineral surfaces (Littlefair et 325   al., 2017), however adsorption onto minerals tends to favour humic-like, oxygen-rich compounds, typically considered recalcitrant, over protein-like compounds (Groeneveld et al., 2020). Thus, sorption could "sort" labile carbon into the dissolved phase and relegate intrinsically recalcitrant ("humic-like", aromatic) carbon to mineral protection as POC. Evidence of this effect includes elevated lability of slump-derived DOM relative to upstream DOM (Littlefair et al., 2017; Littlefair and Tank, 2018), low lability of slump POM (Shakil et al., 2020; this study) and striking compositional similarity of DOC from slump- 330   impacted streams on the Peel Plateau to that from other circumpolar regions with mineral soils (Wologo et al., 2021). However, when particles settle out, anoxia could result in release of adsorbed DOC to the overlying water column (Peter et al., 2016), which may explain the switch from mean DOC decreases to DOC increase in comparisons between rotated and settled slump treatments. Since both selective sorption in the water column and desorption in sediment deposits are likely to happen along the aquatic continuum, it may be difficult to detect or quantify either of these processes *in situ*. Sorption appeared to occur in 335   2019 sterilized bottles as DOC concentrations declined but POC increased, and DOM aromaticity and molecular weight decreased. No change in DOM in unsterilized bottles may reflect sorption (loss of aromaticity) and degradation (gain of aromaticity) acting simultaneously, which further underlines challenges in quantifying process by measuring OC changes in bulk incubations and why DOC declines were not consistently followed by increases or decreases in $SUVA_{254}$ across experiments, even when losses were consistent as in 2016.

340

We note that rapid within-slump processing of labile TOC fractions prior to entrainment within streams may still occur, as supported by high within-slump $NH_4^+$ concentrations (Shakil et al., 2020) indicative of decomposition (Tanski et al., 2017), low representation of labile compounds in the slump scar zone and stream sediments relative to headwall sources (active layer)(Keskitalo et al., 2021), and excess $CO_2$ in within-slump rillwater resulting from both heterotrophic respiration and geogenic production (Zolkos et al., 2019)(Fig. 3f). Past work indicates that OC rapidly lost within-slump may predominantly originate from the active layer (Bröder et al., 2021) and Holocene-age permafrost in areas where organic material buried in colluvial deposits from past slumping has preserved organics (Lacelle et al., 2019). In addition to serving as a possible marker for decomposition, high concentrations of $NH_4^+$ may stimulate nitrification and associated chemosynthetic carbon sequestration. Though we did not see significant TOC gains in our 2019 experiment, ammonia loss coupled with nitrite production suggests active nitrification. Nitrifying bacteria have slow growth rates (Sinha and Annachhatre, 2007; Bock and Wagner, 2013), with the molar ratio of $NH_4^+$ consumed to carbon fixed ranging from $25 - 100$ (Ward 2013). Using this stoichiometry and initial $NH_4^+$ concentrations estimated across 2015-16 incubations (Table S4) indicates that nitrification would be unlikely to fix more than 1 µM of carbon, in comparison to OC gains of $601 \pm 459$ µM (mean $\pm$ 95% CI of SE <20 µm incubation, Fig, 3B, Table S4). Chemolithoautotrophy by sulfur oxidizing bacteria can also sequester carbon (Klatt and Polerecky, 2015) with the ratio of $CO_2$ sequestered to $O_2$ consumed ranging from $0.09 - 0.41$ for aerobic thiosulfate oxidizers (Klatt and Polerecky 2015). The process has been noted to be an important carbon sequestration mechanism in mine tailings (Li et al. 2019). Although the role of aerobic microorganisms in sulfide oxidation is commonly associated with acidic-pH conditions as in Li et al. (2019), this process can also occur at circumneutral pH (Percak-Dennett et al. 2015). Given the high sediment concentration in streams affected by slumping (can exceed 800 g $L^{-1}$, Shakil et al. 2020), and the prevalence of sulfide minerals and oxidation across the Peel Plateau (Zolkos et al., 2018), chemolithoautotrophy associated with sulfide oxidation is a mechanism worth exploring as a counterbalance to OC mineralization. Precise techniques such as isotope labelling (Spona-Friedl et al., 2020) and the tracking of genes associated with carbon fixation processes (Percak-Dennett et al., 2017) may circumvent challenges associated with POC measurement errors and tracking multiple processes acting on OC end-point measurements.

**5 Conclusion**

Permafrost thaw slumping is increasing TOC concentrations in streams across the Peel Plateau (Canada) by orders of magnitude, almost entirely in the form of POC (Kokelj et al., 2021; Shakil et al., 2020; Keskitalo et al., 2021). Across incubations conducted including slump-POC, we found a maximum of 4% of the initial TOC was lost within 27 days and estimate that this would scale to approximately 7% of slump-derived TOC being lost during transport in streams. Changes in DOC and POC fractions suggest that this loss is primarily driven by losses in DOC, with slump-POC displaying low biodegradability across our incubations. This finding, in addition to previous findings that show that the majority of sediment lost from slumps is quickly deposited in debris tongues rather than immediately transported downstream (Kokelj et al., 2021;

Shakil et al., 2020), suggest that the majority of the TOC exiting the slump scar zone is likely to be sequestered in sediments after mobilization to streams. While our experiments examine material exiting rather than within the slump scar zone, thus missing potential within-slump degradation, our estimates highlight that POC degradation rates may be much lower than those for DOC; an important consideration for models of permafrost organic matter degradation, particularly given the dominance of organic matter export in the particulate form from thermokarst features. Our results indicate that even when in suspension, thaw slump-derived POC on the Peel Plateau may be subject to far slower rates of degradation than estimates currently used in models of carbon release from abrupt permafrost thaw (Turetsky et al. 2020) , underlying the need to constrain regional variation in a component of the carbon cycle that has undergone substantial perturbation. Further, increased input of minerals alongside increases in organic carbon into streams creates significant potential for carbon sequestration via abiotic (sorption, mineral protection) and biotic (chemolithotrophy) processes. Targeted investigations of these multiple processes acting simultaneously on carbon dynamics requires specific quantification in landscapes experiencing rapid change.

**Appendices**

**Appendix A: Data analysis details and ANOVA results**

The percent change of OC was used to measure differences in biodegradability:

$$\%\Delta OC = \frac{(OC_{T_n} - OC_{TOaverage})}{OC_{TOaverage}} \tag{A1}$$

In eq. A1 above, $OC_{T_n}$ is the DOC, POC, or TOC measured at an end time point, and $OC_{TOaverage}$ is the mean OC measured at the beginning of the experiment. Since multiple outcomes were tested for ANOVAs, p-values of main tests were corrected for false discovery rate using padjust() from R package "emmeans" (Lenth, 2021). For 2015, main tests were corrected for 19 tests since the SU treatment was tested in 4 comparisons and 3-4 outcomes were tested per comparison. For 2016, main tests were corrected for 4-5 tests since 4-5 outcomes were tested per ANOVA. Follow-up Tukey or Games-Howell adjusted pairwise t-tests were conducted only when an interaction or main test of interest was significant. Prior to PCA we (a) used Pearson correlations to remove variables such that no variables within the PCA had a Pearson correlation greater than 0.9; (b) log-transformed all variables to prevent skew and (c) conducted a detrended correspondence analysis to ensure linearity of the dataset.

**Table A1: Two-way ANOVAs of tests of sources, filtrate, and settling on biodegradability of POC in 2015 experiments. Follow-up Tukey-adjusted pair-wise t-tests are shown where significant interactions were present. Since SU treatments were tested 3 times and multiple outcomes were tested, p-values reported in two-way ANOVAs were corrected for false discovery rate (19 tests). Degrees of freedom associated with treatment, site, interaction, and residuals are 1, 2, 2, and 12 respectively for all tests.**

| Effect | | Treatment | | | | Site | | Treatment*Site | |
|---|---|---|---|---|---|---|---|---|---|
| Test | Variable | estimate | error | F/t[a] | p | F | p | F | p |
| Control | $\Delta SUVA_{254}$ | - | - | 1.45 | 0.48 | 1.64 | 0.37 | 1.42 | 0.66 |
| (SU-UF) | %$\Delta$DOC | - | - | 0.54 | 0.75 | 0.02 | 0.98 | 0.54 | 0.81 |
| | %$\Delta$TOC | - | - | 0.49 | 0.75 | 0.07 | 0.98 | 0.98 | 0.80 |
| | ln(k) | - | - | 305.57 | **0.00** | 6.43 | **0.04** | 8.92 | **0.04** |
| | HA | 2.01 | 0.17 | 11.80 | **<<.001** | - | - | - | - |
| | HB | 2.02 | 0.17 | 11.83 | **<<.001** | - | - | - | - |
| | HD | 1.13 | 0.17 | 6.64 | **<<.001** | - | - | - | - |
| Source | $\Delta SUVA_{254}$ | | | 2.22 | 0.34 | 0.14 | 0.98 | 6.51 | 0.08 |
| (SU-UU) | HA | *0.40* | *0.11* | *3.80* | 0.00 | - | - | - | - |
| | HB | *-0.06* | *0.11* | *-0.52* | 0.61 | - | - | - | - |
| | HD | *-0.07* | *0.11* | *-0.70* | 0.50 | - | - | - | - |
| | %$\Delta$DOC | -7.06 | 2.46 | 8.22 | 0.05 | 1.13 | 0.47 | 1.53 | 0.66 |
| | log(%POC+50) | -0.19 | 0.04 | 27.40 | **0.00** | 5.08 | 0.07 | 2.39 | 0.42 |
| | %$\Delta$TOC | - | - | 2.22 | 0.34 | 1.73 | 0.37 | 0.78 | 0.80 |
| | ln(k) | | | 172.43 | **0.00** | 43.21 | **0.00** | 67.84 | **0.00** |
| | HA | *1.69* | *0.15* | *11.60* | **<<.001** | - | | - | |
| | HB | *1.90* | *0.15* | *13.04* | **<<.001** | - | | - | |
| | HD | *-0.28* | *0.15* | *-1.89* | **0.08** | - | | - | |
| Filtrate | $\Delta SUVA_{254}$ | - | - | 2.52 | 0.34 | 11.81 | **0.01** | 0.41 | 0.81 |
| (SD-SU) | %$\Delta$DOC | - | - | 0.05 | 0.87 | 2.71 | 0.21 | 2.90 | 0.42 |
| | %$\Delta$POC | - | - | 0.11 | 0.85 | 2.66 | 0.21 | 0.20 | 0.89 |
| | %$\Delta$TOC | - | - | 0.37 | 0.75 | 2.66 | 0.21 | 0.72 | 0.80 |
| | k | - | - | 0.24 | 0.81 | 20.12 | **0.00** | 0.19 | 0.88 |
| Settling | $\Delta SUVA_{254}$ | - | - | 0.10 | 0.85 | 10.46 | **0.01** | 0.77 | 0.80 |
| (SS-SU) | %$\Delta$DOC | - | - | 5.84 | 0.10 | 0.11 | 0.98 | 0.39 | 0.81 |
| | %$\Delta$POC | - | - | 0.38 | 0.75 | 1.08 | 0.47 | 0.21 | 0.88 |
| | %$\Delta$TOC | - | - | 0.00 | 0.96 | 1.36 | 0.43 | 0.43 | 0.81 |
| | ln(k) | -0.67 | 0.07 | 81.68 | **0.00** | 11.47 | **0.01** | 2.46 | 0.42 |

[a]F-values are reported for two-way ANOVAS, t-values are reported for follow-up pairwise t-tests

**Table A2: Welch's ANOVA and follow-up Games-Howell pair-wise t-tests of differences between size fractions (2016 experiment). P-values for main ANOVAs are adjusted for false-discovery rate to account for multiples outcomes tested (5 tests). Follow-up tests were only conducted when main ANOVA tests showed a significant difference. Size fractions: SN = 63 – 2000 µm, SL= 20 – 63 µm, SMSC = <20 µm.**

| Variable | F | $df_{num}$ | $df_{denom}$ | p | Follow-up tests | | | | |
| --- | --- | --- | --- | --- | --- | --- | --- | --- | --- |
| | | | | | Estimate | 95% error | t | df | p |
| $\Delta SUVA_{254}$[a] | 41.63 | 2 | 5.8 | **0.001** | | | | | |
| *SL vs SN* | | | | | 2.00E-02 | 1.75E-01 | 0.46 | 5.2 | 0.894 |
| *SMSC vs SN* | | | | | 3.80E-01 | 1.75E-01 | 7.14 | 5.0 | **0.002** |
| *SMSC vs SL* | | | | | 3.60E-01 | 2.60E-01 | 8.62 | 6.0 | **<0.001** |
| %$\Delta$DOC | 22.21 | 2 | 4.9 | **0.007** | | | | | |
| *SL vs SN* | | | | | -6.09E+00 | 1.97E+01 | 2.00 | 4.8 | 0.200 |
| *SMSC vs SN* | | | | | -1.45E+01 | 2.00E+01 | 5.40 | 3.4 | **0.018** |
| *SMSC vs SL* | | | | | -8.39E+00 | 1.12E+01 | 5.20 | 4.3 | **0.012** |
| %$\Delta$POC | 2.81 | 2 | 4.5 | 0.161 | | | | | |
| %$\Delta$TOC | 5.13 | 2 | 4.4 | 0.089 | | | | | |
| ln(k) | 66.66 | 2 | 5.9 | **<0.001** | | | | | |
| *SL vs SN* | | | | | -4.34E-05 | 4.01E-03 | -0.01 | 9.0 | 0.992 |
| *SMSC vs SN* | | | | | 3.87E-02 | 4.01E-03 | 9.66 | 9.0 | **<<0.001** |
| *SMSC vs SL* | | | | | 3.87E-02 | 4.01E-03 | 9.67 | 9.0 | **<<0.001** |

410

**Table A3: Two-way ANOVAs for transect experiment examining effects of filtrations (unfiltered vs. filtered) and distance (immediately downstream vs. 2.79 km downstream) from slump SE and interactions between the two. Since multiple (4-5) parameters from the same experiment are tested, p-values were corrected for false discovery rate. Degrees of freedom associated with distance, filtration, interaction, and residuals are 1, 1, 1, and 12 respectively for all tests.**

| Variable | Distance | | Filtration | | Distance*Filtration | | Follow-up tests | | | | |
|---|---|---|---|---|---|---|---|---|---|---|---|
| | F | p | F | p | F | p | Est. | se | t | df | p |
| ΔSUVA254 | 0.21 | 0.659 | 5.59 | 0.048 | 12.44 | 0.013 | | | | | |
| DN vs 2.79 km DN (filtered) | | | | | | | -0.09 | 0.04 | -2.174 | 12 | 0.050 |
| DN vs 2.79 km DN (unfiltered) | | | | | | | 0.12 | 0.04 | 2.815 | 12 | 0.016 |
| %ΔDOC | 7.39 | 0.031 | 17.00 | 0.003 | 10.87 | 0.013 | | | | | |
| DN vs 2.79 km DN (filtered) | | | | | | | 0.67 | 1.64 | 0.409 | 12 | 0.689 |
| DN vs 2.79 km DN (unfiltered) | | | | | | | -6.96 | 1.64 | -4.253 | 12 | 0.001 |
| %ΔPOC[a] | 12.01 | 0.031 | - | - | - | - | | | | | |
| %ΔTOC | 1.71 | 0.269 | 1.99 | 0.184 | 0.00 | 0.949 | | | | | |
| ln(k) | 12.65 | 0.020 | 55.96 | 0.000 | 0.95 | 0.466 | | | | | |

[a]Percent change in POC not tested for filtration or interaction effect because negligible POC concentrations in filtered treatments

## Appendix B: Characterization of each size fraction

Size fractions with smaller particles sizes had lower organic matter content (lower percent organic carbon) and associated organic matter had a greater $^{14}$C age (Table B1). Extraction of size fractions was conducted in triplicate, but one replicate of the size fraction > 63 µm had to be removed due to optical density concerns (supplementary S3.8). A biplot of PCA components 1 and 2 that explained 92% of the variation in optical indices of BEPOM revealed that extractions of organic matter in the clay and silt size fractions had a greater relative contribution of UVA humic-like peak C (Fig. B1), which has been characterized as terrigenous organic matter that has undergone less chemical reworking than peak A (Stubbins et al. 2014). While peak C was not negatively correlated with peak A, it was strongly correlated with total absorbance per unit of sediment extracted (Table S6).

**Table B1: Characteristics of size fractions used in 2016 experiment. %POC indicates POC: TSS. sd = standard deviation. SEM = standard error of the mean.**

| Size Fraction Category | Unfractionated | 0.063 - 2mm | 0.020 - 0.063 mm | <0.020 mm |
|---|---|---|---|---|
| Mean particle size | 11.9 | 48.9 | 21.67 | 5.78 |
| sd | 3.78 | 4.6 | 2.9 | 2.79 |
| F$^{14}$C | NA | 0.4113 | 0.1332 | 0.056 |
| sd | NA | 0.0019 | 0.0012 | 0.0006 |
| calBP (percent representation) | NA | 8020 - 7926 (88.3%) | 19791 - 19292 (95.4%) | 27640 - 27250 (95.4%) |
| | | 7896 - 7871 (7.1%) | | |
| %POC | 1.6 | 3.12 | 1.35 | 1.67 |
| SEM | 0.03 | 0.09 | 0.02 | 0.02 |
| % of sum initial POC across fractions | NA | 12% | 15% | 73% |

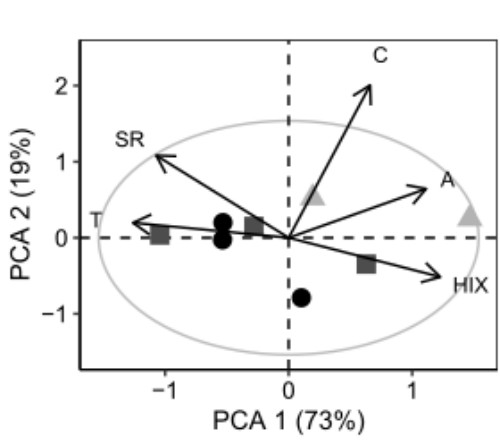

**Figure B1.** Principal components analysis of optical indices for BEPOM of different size fractions. PCA is shown in scaling 1. The grey circle marks the circle of equilibrium contribution. Abbreviations of optical indices are as in Table S5; SN = 63 – 2000 µm, SL= 20 – 63 µm, SMSC = <20 µm.

**Appendix C: Absolute changes in SUVA$_{254}$**

**Table C1: Absolute changes in SUVA$_{254}$ in 2015-16 experiments.**

| Year | Treatment | Site | $\Delta$SUVA$_{254}$ | 95% error | n |
|------|-----------|------|----------------------|-----------|---|
| 2015 | Filtered upstream (UF) | HA | 0.08 | 0.36 | 3 |
|      |           | HB | 0.11 | 0.70 | 3 |
|      |           | HD | 0.02 | 0.08 | 3 |
|      | Unfiltered upstream (UU) | HA | -0.25 | 0.20 | 3 |
|      |           | HB | -0.06 | 0.35 | 3 |
|      |           | HD | -0.02 | 0.44 | 3 |
|      | Slump in filtered upstream (SU) | HA | 0.16 | 0.47 | 3 |
|      |           | HB | -0.11 | 0.16 | 3 |
|      |           | HD | -0.09 | 0.15 | 3 |
|      | Slump in filtered downstream (SD) | HA | 0.20 | 0.28 | 3 |
|      |           | HB | -0.07 | 0.06 | 3 |
|      |           | HD | 0.04 | 0.12 | 3 |
|      | SU settle (SS) | HA | 0.19 | 0.41 | 3 |
|      |           | HB | -0.05 | 0.20 | 3 |
|      |           | HD | -0.20 | 0.60 | 2 |
| 2016 | Unfractionated | SE | 0.46 | 0.41 | 4 |
|      | 2 - 0.063 mm | SE | 0.06 | 0.15 | 4 |
|      | 0.063 - 0.020 mm | SE | 0.09 | 0.10 | 4 |
|      | < 0.020 mm | SE | 0.45 | 0.09 | 4 |
|      | Upstream | SE | -0.07 | 0.11 | 4 |
|      | Upstream filtered control | SE | -0.07 | 0.07 | 4 |
|      | Downstream | SE | -0.10 | 0.17 | 4 |
|      | Downstream filtered control | SE | 0.07 | 0.05 | 4 |
|      | 2.79k Downstream | S2 | -0.01 | 0.03 | 4 |
|      | 2.79k Downstream filtered control | S2 | -0.05 | 0.07 | 4 |

**Appendix D: Flow chart of 2019 experiment**

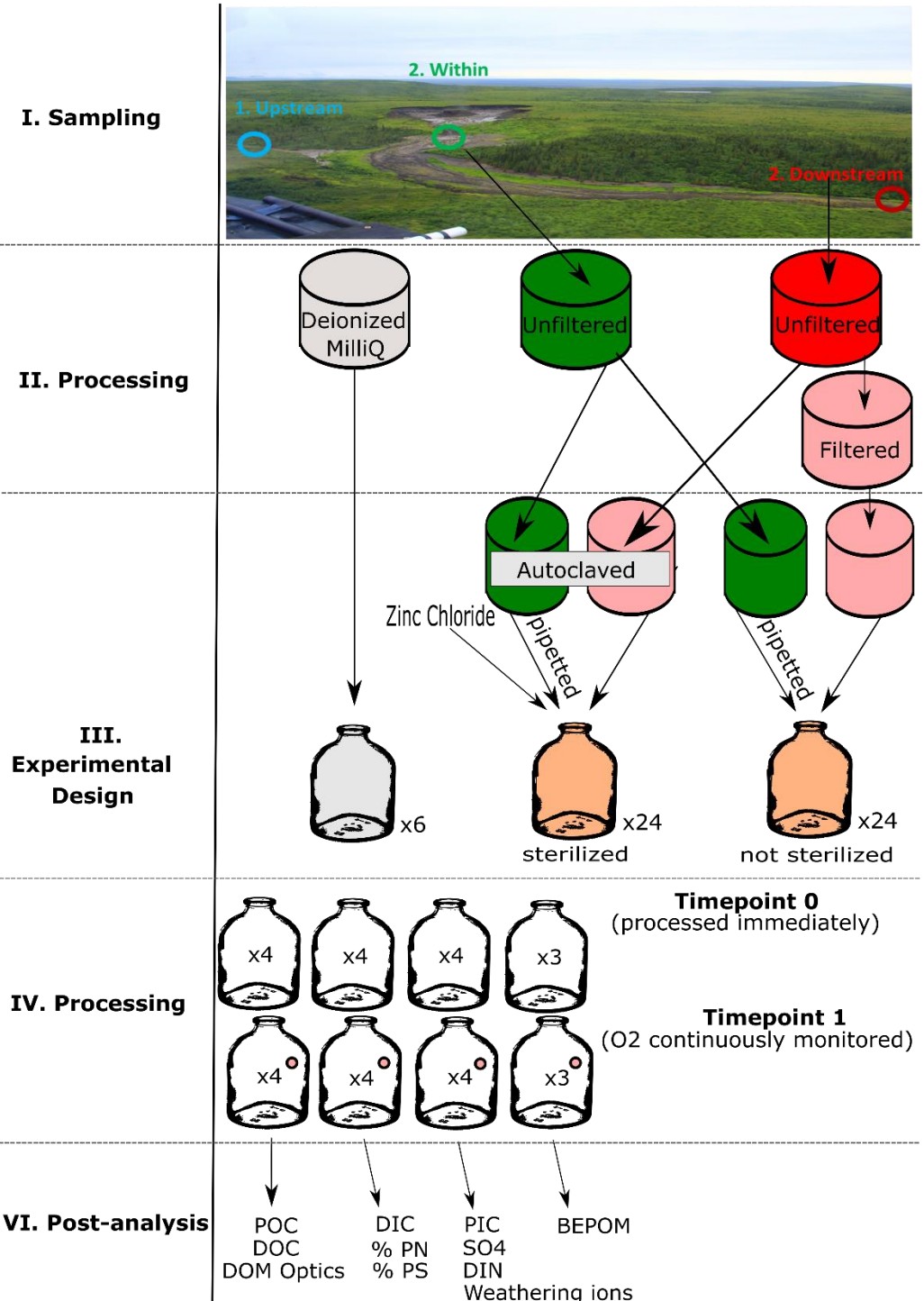

**Figure D1.** Flow chart for processing of 2019 experiment. Note, that only 2 bottle replicates were used to assess changes in BEPOM for sterilized treatments. Analyses show particulate (POC) and dissolved (DOC) organic carbon, dissolved organic matter (DOM) optics (absorbance and fluorescence), dissolved (DIC) and particulate (PIC) inorganic carbon, % particulate nitrogen (PN) and sulfur (PS), dissolved inorganic nitrogen (DIN), $SO_4^{2-}$, weathering ions, and base-extracted particulate organic matter (BEPOM).

**Code Availability**

Code for this manuscript is available upon request.

**Data Availability**

Data is openly available through the Polar Data Catalogue (CCIN reference #: 13237). (Shakil et al. 2021)

**Supplement link**

A supplementary pdf file has been uploaded for review with this manuscript.

**Author contribution**

SS and SET led the design of the study. SS led data collection, data analysis and interpretation, and manuscript writing. JEV contributed to the initial development of the idea, and JEV and SZ contributed to study design and data interpretation. SZ

contributed to laboratory methods for data collection. All authors contributed to manuscript writing.

**Competing interests**

The authors declare they have no conflict of interest.

**Acknowledgements**

This work took place within the Gwich'in Settlement Region, and we are thankful for support from the Tetlit Gwich'in

Renewable Resources Council and Western Arctic Research Centre. We are further thankful for the field assistance of Christine Firth, Elizabeth Jerome, Andrew Koe, Joyce Kendon, Maya Guttman, Luke Gjini, and Lindsay Stephen. Maya Guttman and Joyce Kendon also helped experiment set up and sample processing. Hailey Verbonac assisted with $O_2$ measurements during our 2015 experiments conducted in Inuvik. This manuscript also benefitted from helpful discussions with: Steve Kokelj with regards to field sampling and perspectives on landscape changes in the region; Matthias Koschorreck and Rafael Marcé with

regards to chemoautotrophic processes; and Alex Wolfe who first provided advice to broaden consideration of what affects oxygen and carbon dynamics. The rotator used for incubations was designed and manufactured by Technical services staff in the Department of Mechanical Engineering at the University of Alberta, supervised by Roger Marchand. Funding for this study was provided by the Natural Sciences and Engineering Research Council (NSERC), Polar Continental Shelf Program (Natural

Resources Canada), Campus Alberta Innovates Program, ArcticNet, CiCan Cleantech Internship Program, Environment

Canada Science Youth Horizons Internship, Northern Scientific Training Program, University of Alberta and UAlberta North, and the Aurora Research Institute. Personal support to SS was provided by NSERC and the Garfield Weston Foundation. Research for this paper was conducted under NWT research licences: 15685 (2015); 15685 (2016); 15887 (2017); 16575 (2019).

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
