# Peer review of "Low biodegradability of particulate organic carbon mobilized from thaw slumps on the Peel Plateau, NT, and possible chemosynthesis and sorption effects"

_Biogeosciences, 2021_

## Author Comment (AC1)

**Reviewer 1**

**General Comments:**
This paper advanced knowledge regarding mobilized POC biodegradability as a result of Arctic thaw slumps. The identification of the rate of biodegradability of slump-mobilized POC answers an important piece of the lateral carbon flux puzzle in this region, making this paper very worthy of publication. Ultimately, this paper also answers the separate question by proxy, that there is a trend for slump-mobilized DOC to decrease during incubation despite the low biodegradability of POC and TOC. The knowledge gap is clearly stated, and the introduction does a comprehensive job of outlining the main question. The paper also detailed comprehensive experimentation over the course of many years in order to answer a series of related, nested questions. There are a few modifications and clarifications that I have outlined below that I believe would help to heighten the considerable impact of this paper's findings. I have broken up my main points into four bullets below. Line edits and more detailed questions follow in the Specific Comments and Technical Comments sections.

We would like to thank the reviewer for their careful comments, and also for this positive assessment of the manuscript.

- Relative to what the degradation of organic carbon would have been had it remained frozen in the Arctic tundra, oxidative loss of 4% POC per month could be significant. At this rate, this accounts for a potential loss of 16% over the course of a 4-month growing season. When scaled up across the Arctic or scaled up over many years, this represents a considerable C degradation pathway. I believe it is important for the author to contrast this 4% loss with the alternative, had POC not been mobilized by thaw. For instance, in the absence of thaw slumps mobilizing this POC, can we assume negligible loss of permafrost organic carbon over the same timescale? The paper's tone does not do this impactful result justice. I would re-casting the significance for C cycling in contrast to the degradation rate in the absence of slump-induced C mobilization. Though the biodegradability may be low, it is still quite important at the rate of 4% per month.
Thank you for the comments. We would like to clarify a few points.
  (1) The 4% loss is really a measure of TOC loss and we have clarified this at relevant points in the manuscript. This the highest degradation rate we measured for TOC and we believe it is primarily due to DOC loss (now clarified in the abstract). In many experiments we did not see any significant loss of TOC, and in some cases a gain.
  (2) Furthermore, this 4% loss rate is what is measured under oxygenated conditions with particles maintained in suspension. Thaw slumps within our study region mobilize material far in excess of what the stream can transport (Shakil et al. 2020; Kokelj et al. 2021) which will substantially reduce the ability of organic carbon to oxidize this material.
  (3) It is unlikely that the oxidation of TOC scales linearly across time. Degradation rates tend to decrease over the course of an incubation, typically resulting in a "levellingoff" of the carbon dioxide produced or the organic carbon consumed (see, for example, the log-associated calculation of k as a reaction rate coefficient, for many studies on decomposition). We were unable to quantify this change in rates in our experiments because we used end-point organic carbon measurements. We could not use our high-resolution oxygen measurements to quantify this as a proxy since we believed that processes other than organic carbon degradation were significantly consuming oxygen (as highlighted in the paper). Thus we used our best proxy of the most similar experiment conducted in the most similar terrain (the work by Tanski et al. 2019, which was conducted just north of our study site, on the similarly glacially conditions Herschel Island) to determine our estimate of 7% of TOC mineralized during riverine transport.

(4) We highlight, now with edits, "These findings are consistent with generally lower $CO_2$ fluxes and concentrations downstream relative to upstream of slumps (Zolkos et al. 2018, Zolkos et al. 2019), despite orders of magnitude increases in POC and TOC (Shakil et al. 2020)."

(5) We feel that using 0% mineralization as a point of reference may be a bit absolute considering that some microbes can adapt to mineralize organic carbon within permafrost (Leewis et al. 2020) and the large losses of $CO_2$ that can occur during the winter in the Arctic (Natali et al. 2019). However, we do now note that this mineralization may be elevated relative to mineralization occurring within the frozen conditions of permafrost but also note that our estimate is considerably lower than the estimate used by Turetsky et al. (2020) that 2/3rds of DOC/POC from hillslope abrupt thaw will be mineralized and highlights a need for region specific estimates of permafrost carbon release (also added to address comments from Reviewer 2)

- This paper includes many great experiments; however, the conclusion is a little truncated. Echoing point 1, the conclusion is a good place to reiterate the significance of the main finding, of 4% POC loss per month.
  We don't believe stating a loss of 4% POC per month is accurate, please see comment above. We have now modified the conclusion (with some edits in response to comments from Reviewer 2) to add more detail regarding our findings and their relevance to permafrost carbon release.

- Further, I would suggest taking a stronger stand on each of your experiments and the text of the discussion, where you parse out the relative importance of each of the potential POC sequestration pathways (abiotic and biotic). The conclusion would be improved if it included more regarding the vulnerability of this mobilized and subsequently sequestered C. Will the sequestered C be vulnerable to a faster rate of decomposition? What is the most important sequestration mechanism in the second to last sentence (L308)?
  Thanks for these comments. We dedicate a paragraph in the discussion to discuss pathways of chemolithotrophy and a paragraph to discuss sorption, the effects of sorption on DOC biodegradability, and potential conditions in the stream system where adsorbed DOC could be released. We do not feel we have enough supporting evidence quantifying

these other two processes to make firmer statements than contained in our original text and also don't feel we have any basis to estimate the rate of decomposition of sequestered carbon. Our purpose was to highlight that these processes may be significant in typical experimental designs used to quantify degradation rates and need to be accounted for and to encourage future research to explore these avenues.

- The supplemental flow charts (S5-S7) are incredibly useful in aiding the reader's understanding of the experimental design. Within the text itself, the location/code of individual slumps and the treatment codes distract the reader from the main findings beyond Figure 1, which orients the reader to each slump location and code. In general, I am curious why the authors did not combine the slumps that were similar in their analyses, treating them as replicates of one another (with the exception of the slump with some encroachment reported). As a general suggestion overall, if it is possible to remove all reference to specific site acronyms (rather, refer to each site as site 1, 2, 3, etc.) and to refer to the treatments with complete description, e.g., "unfiltered upstream" rather than by acronym "UU", I believe the text clarity and readability would be greatly improved. This is a minor update that I believe would have a major impact.

(1) Thank you for the comment on the flow diagrams. We are glad to hear that the reviewer found these to be useful. We are happy to move the flow diagrams associated with the 2015 and 2016 experiment to the main text (or appendix associated with the main text, also in response to comments from Reviewer 2). Since our 2019 experiment was simple and really a replication of the SD treatment in 2015, with an added sterilization, we think it's best to keep this diagram in the appendix for brevity. With respect to slump codes, we have chosen to retain these as-is because these sites have been the subject of multiple investigations. As a result, we would like to provide readers with the tools (i.e., clear reference to existing slump codes) to compare our results with those from previously published studies. Although we acknowledge that the treatment codes do add some complexity to the manuscript, we provide Table 1 and Figures S5-S7 (which can be moved to the main text / appendix) as a quick reference guide to help alleviate this concern.

(2) There was large variation in rates of oxygen consumption between slump-affected treatments between sites (Figure 2 a-c). We believe this is largely associated with large variations in sediment concentration which are difficult to control (Table S2). We wanted to keep the slump sites separate so that the differences between treatments wasn't lost within the variation between sites and it also is a truer representation of our experimental design.

(3) In the text we do primarily use the text description to refer to treatments but kept the acronyms in brackets for ease of reference to figures since there is not enough space in the figures to put the complete text description. With movement of the flow charts to the main text we hope that these acronyms will be easier to follow.

- <1 week incubation times seem very short. For soil incubations, this short time would be considered a disturbance measurement since there are artifacts from handling and setting

up an experiment in conditions away from the field. Furthermore, POC from older carbon permafrost soil (as evidenced by radiocarbon age) would likely have a slow turnover time, which by nature takes longer to measure rates. Please add some visible text to the discussion and conclusion that the short term experiments might be limited in both detecting the actual rate of change (Type II statistical error), and the role that the novel lab conditions may play during this time period.

(1) The experiments range from 7 to 27 days. However, while these time frames may be short for soil experiments they are not uncommon in the aquatic literature (Vonk et al. 2015, Richardson et al. 2013). In Tanski et al. (2019), the experiment we compare to, ~58% of the total $CO_2$ production occurred within 27 days. In Spencer et al. (2015), a study conducted in a region where high biolability rates have been noted for DOC, they found ~50% DOC loss in <7 days). Thus, we have used these studies to frame the timespan of our incubation, and also chose to err on the side of shorter incubations to avoid issues related to "bottle effects" (e.g., Vonk et al. 2015).  Further, we do have text (line 256 – 265, "Our findings of low POC biodegradability is likely conservative…") discussing the short time frame of our experiments and estimating what the percent loss would be if we extrapolated beyond the time frame used (see also responses above).

(2) Older radiocarbon age doesn't always equate to slower turnover time for permafrost soils. Cryoturbation, low temperatures, and poor drainage over long time periods can result in carbon becoming sequestered in frozen soils with minimal exposure to degradation, although this varies across landscapes based on permafrost history, geology, etc. (Tank et al. 2020). This is highlighted in papers that have noted relatively higher biodegradability for relatively older carbon via experiments or lipid proxies (e.g., Spencer et al. 2015, Bröder et al. 2020).

**Specific Comments:**

This paper is primarily focused on POC not POC and DOC, however, the ultimate findings suggest that POC fractions studied have lower biodegradability than DOC and I believe that contrasting the two broader size classes of stream OC and how they may interact could be of use given the ultimate findings (e.g., increased POC mineral input into streams has the potential to increase DOC sorption).

In the last section of the discussion (lines 289 – 305) we contrast the differences between POC and DOC, highlight interactions, but also highlight complications in quantifying different mechanisms with these experiments. We have adjusted the abstract to highlight that most of the TOC loss appears to be due to loss from the DOC pool and have also added in further text in the discussion to contrast previous findings on lability and sources of the two pools.

I also believe that POC and DOC transport is an important aspect of lateral carbon fluxes worthy of mentioning early on in the abstract, albeit briefly. Transport of carbon is the initial mechanism that allows for mineralization into CO2 and re-sequestration into sediments.

We modified a sentence in the abstract and introduction to make it clear that POC is the primary mode through which organic carbon is mobilized to streams in landscapes affected by hillslope thermokarst disturbance.

Abstract:
"Warming and wetting in the western Canadian Arctic is accelerating thaw-driven mass wasting by permafrost thaw slumps, increasing total organic carbon delivered to headwater streams by orders of magnitude due to orders of magnitude increases in particulate organic carbon (POC)."

Introduction:
"Thaw slumping along stream sites in this region can increase TOC yields by orders of magnitude almost entirely due to orders of magnitude increases in POC yields (Shakil et al. 2020)."

L11: Mineralization as CO2 and sedimentation are two POC fates, but this sentence does not address re-sequestration of stream C by aquatic plants or transportation downstream (though transportation is not an ultimate, chemical fate). I believe 1) it would be useful if the abstract jumped right into POC as this is the primary focus of the paper's research OR 2) for the abstract to include mention of transportation as the mechanism allowing for soil organic carbon to become transported POC, mineralized CO2, or re-sequestered sediment within stream systems.

Suggestion 1: "Upon thaw, permafrost particulate organic carbon (POC) may be mineralized into CO2…"
Suggestion 2: "Upon thaw, permafrost carbon entering and transported within streams may be…"

Following suggestion 2 we have added "and transported within".

L30-35: Transport is covered in this section, I believe it should be mentioned in the abstract, briefly as is presented in Specific Comment #1 above. The dichotomy of fates as it relates to the transport trajectory (transport vs deposition according to size and density fractions) is ultimately relevant to the study findings.

We incorporated suggestion 2 for the comment above to address this.

L32: It is probably worth mentioning that anoxia reduces overall mineralization rates but also shifts carbon loss towards methane (Schaedel et al. 2017 Nature Climate Change)

We have incorporated this into the sentence:
"…(Peter et al. 2016), though carbon release can shift to be in the form of methane (Schaedel et al. 2017)."

L40: Might be helpful to discuss different sources of POC in slump affected- and non-affected streams so reader can understand why lability might go up/down.

We have modified the sentence to read,
"Slump-POC chemical composition suggests lower bioavailability as POC sources shift from the active layer and some periphyton material to Pleistocene-aged organic carbon and petrogenic organic carbon mobilized from permafrost (Shakil et al. 2020, Bröder et al. 2021). However, POC bioavailability has not been experimentally assessed."

L127: circumneutral-pH, in my experience, pH of many Arctic water tracts is closer to pH5 than pH7.

We note that pH in the streams in this study can be quite variable but tend to be circumneutral, often varying around pH7 and most ranging from pH 6 – 8 (see supplementary data of Shakil et al. 2020). This text has now been added.

L190: for clarity, identifying SE particles as slump SE would be useful and parallel HA slump particles later in the sentence. However, see point #3 in the general comments.

We have added the term "slump".

L184: most organic matter is partially oxidized because it has oxygen molecules. For example, glucose has a lot of oxygen molecules. Would this line be expected to be a 1:2 line instead of a 1:1 line? Most organic matter has oxygen as a part of it, does this change the heterotopic respiration line of 1:1?

Thanks for this comment. The net balance of the aerobic respiration of glucose is represented as:

$$C_6H_{12}O_6 + 6O_2 \rightarrow 6CO_2 + 6H_2O,$$

thus, the complete aerobic oxidation of glucose is results in a respiratory quotient of 1, (on balance, the $CO_2$ produced is equivalent to the moles of $O_2$ consumed). This ratio has been documented for the aerobic respiration of glucose in environmental work in soils and litter (e.g., Dilly et al. 2001). Although compounds oxidized in aquatic systems certainly vary beyond glucose (as noted in Berggren et al. 2012) and thus the respiratory

quotient for bulk organic carbon can vary, we chose to represent a 1:1 line given the long history of an assumed respiratory quotient of 1.0 for heterotrophic respiration in aquatic systems, and measured heterotrophic RQ values close to this value in boreal systems (e.g., see Berggren et al. 2012). Further, we note that this doesn't impact our commentary with reference to the figure since even a 1:2 line would still be trending in a different direction that our data.

L250-255: Some DOC may be decreasing as it is converted to $CO_2$ alongside consumption of $O_2$ as shown in Figure 3F and mentioned in L215. I would propose DOC declines as a possible reason for $O_2$ consumption mentioned in L250-255.

Treatments containing particles always had elevated oxygen consumption rates relative to filtered controls for both upstream and downstream water so DOC mineralization can't account for the elevated oxygen consumption. We have added in a sentence to clarify this. Further, we note that our abiotic experiment shows high potential for abiotic consumption of oxygen.

"Despite a lack of TOC or POC loss, oxygen consumption rates in treatments containing particles were always elevated relative to their DOC controls, highlighting that oxygen consumption could not be solely accounted for by DOC mineralization."

Figure 4: Please note, MQ water has been found to carry a baseline amount of DOC, typically below the standard detection limits of a TICTOC but enough to impact radiocarbon measurements (0.5 ppm) if MQ water is used to generated standards.

The Milli-Q water we use has a carbon filter and is quality controlled to be less than 10 ppb TOC so we assumed it negligible compared to the amount of carbon that may be introduced during the experimental set-up. We have added in this detail in section 2.2.4,

"MQ water was sourced from a machine with a carbon filter and was quality controlled to have less than 10 ppb TOC."

**Supplemental Information:**

Page 3: Do you suspect that the varying incubation timing (7, 11, 8, and 27 days) has any impact on the resulting POC degradation?

It's possible, but it is difficult to disentangle the effect of variation in time with variation in experimental set-up and slump sites We have added a note on this in the discussion.

However, we think the gains detected in organic carbon, even within the shorter incubation times of 7, 8, and 11 days (Table S4) may still highlight an important carbon sequestration process that should be further investigated. Though experimental duration

of 7 days is short for soil experiments they are common within the aquatic sciences (see responses above and Vonk et al. 2015)

Unresolved general question: How did you store your samples prior to analysis? How many days were they stored once collected, were they refrigerated, frozen, or acidified? Were they stored in the dark? These questions impact the ultimate degradation of the C within the samples.

Much of this was addressed in the supplementary but we have modified and moved text to the main text for clarity as noted below:

(1) For samples used for the experiment set-up:
Now in section 2.1 (region and field sampling) in the main text – "All samples were processed (i.e., filtered) within 24 hours of collection, apart from within-slump and downstream samples used for adding particles to "unfiltered" treatments in 2016 that were stored in the dark at 4°C until the start of the experiment. Experiments were started within 24 (2015, 2019) – 48 (2016) hours after processing (thus 48 – 72 hours after field collection, details per experiment below). The extra hold time for the 2016 experiment was due to the extra time needed for size fractionation of samples (see below and supplementary S2)."

(2) For samples stored for analysis after removing from the experiment details are in supplementary S3 for each analyte. If missing, details on storage until analysis were added. Storage and analyses of sensitive analytes (e.g., DIN) were conducted following the tested methods of the Canadian Association of Laboratory Accreditation (CALA)-certified Biogeochemical Analytical Service laboratory.

**Technical Corrections:**

Table 1: In my copy of the manuscript, Table 1 text is too large for the cells, with overhanging letters in the first four columns.

We have altered Table 1 to be landscape format.

L56: 1) removing the slump site identifiers entirely from the text regarding the 2016 and 2019 experiments or 2) Identifying which three slump sites were used (HA, HB, HD) in 2015 would be useful for the reader and would mirror the identification of slumps SE and FM3 in the 2016 and 2019 experiments in line 61 and 62, respectively (see comment on L59, below). However, see point #3 in the general comments.

On line 56 we have added "…. three slump sites (HA, HB, HD)..." to specify the sites used in 2015.

L59: Site HD-UP is introduced in the text before the reader is oriented to what site "HD" represents; supplemental Figure S5 does not portray HD-UP, I believe this should be updated to Figure S4 and HD could be introduced in Line 56 as mentioned above. However, see point #3 in the general comments.

With the addition of the slump names for 2015 (see response above) we think this will be more clear since we introduce the UP, IN, DN acronyms in lines 58-59. We also adjusted the Figure 1 caption to indicate that HA, HB, HD, SE, and FM3 are slump sites. This note was here for later reference of why HD upstream treatments were more similar to slump sites than HA and HB (line 153).

Figure 1: It would be beneficial to the reader to identify slump SE on the larger map as well as in the map inset (slumps HB, HA, FM3, and HD are all identified on the larger map, but SE is missing).

The transect for SE was highlighted on the main map. We have now added in a red dot (similar to the other slump sites) on the main map for the location of SE.

Indicating that SE, HB, HA, FM3, and HD are slumps on the map key would be useful.

A notation in Figure 1 has been added.

Within the inset, SE-IN is identified. Should UP, DN-1, and DN-2 also be described with the SE- prefix in the inset?

Inset b of Figure 1 has been modified to explain the UP, IN, DN abbreviations (upstream, within, downstream).

I would suggest labeling the entire inset as the slump SE transect and omitting the label SE- from the IN location. However, see point #3 in the general comments.

We have labelled the entire inset as the 2016 transect experiment.

L81-83: The settling component of the 2015 experiment is distinct from the 2015 incubation experiment. I believe this would be best organized in a subsection, rather than grouping the incubation and settling together by year in one paragraph, as variation over year is not a factor of interest in the overall paper.

Thank you for this comment. We did consider this, however, we wanted to maintain results within each experimental set-up since each experimental had a different design for different goals (as summarized in Table 1). The results for settling are already presented

in a separate subsection ("3.2 Experiment 2015: Effects of background dissolved constituents and settling"). The years are presented for organization purposes, matching with field collection, and so it's clear that the experiments were conducted in different years.

L85: Slump SE is referred to in this section however the 2015 sites were not mentioned by name in the previous section (2.2.1). I'd recommend consistency between the sections. However, see point #3 in the general comments.

We have addressed this, see comments above.

**References**

Berggren, M., Lapierre, J.-F., & del Giorgio, P. A. (2012). Magnitude and regulation of bacterioplankton respiratory quotient across freshwater environmental gradients. *The ISME Journal*, *6*(5), 984–993. https://doi.org/10.1038/ismej.2011.157

Bröder, L., Davydova, A., Davydov, S., Zimov, N., Haghipour, N., Eglinton, T. I., & Vonk, J. E. (2020). Particulate Organic Matter Dynamics in a Permafrost Headwater Stream and the Kolyma River Mainstem. *Journal of Geophysical Research: Biogeosciences*, *125*(2), e2019JG005511. https://doi.org/10.1029/2019JG005511

Bröder, L., Keskitalo, K., Zolkos, S., Shakil, S., Tank, S. E., Kokelj, S. V., Tesi, T., Dongen, B. E. V., Haghipour, N., Eglinton, T. I., & Vonk, J. E. (2021). Preferential export of permafrost-derived organic matter as retrogressive thaw slumping intensifies. *Environmental Research Letters*, *16*(5), 054059. https://doi.org/10.1088/1748-9326/abee4b

Dilly O 2001 Microbial respiratory quotient during basal metabolism and after glucose amendment in soils and litter *Soil Biology and Biochemistry* **33** 117–27

Kokelj, S. V., Kokoszka, J., van der Sluijs, J., Rudy, A. C. A., Tunnicliffe, J., Shakil, S., Tank, S. E., & Zolkos, S. (2021). Thaw-driven mass wasting couples slopes with downstream systems, and effects propagate through Arctic drainage networks. *The Cryosphere*, *15*(7), 3059–3081. https://doi.org/10.5194/tc-15-3059-2021

Leewis, M.-C., Berlemont, R., Podgorski, D. C., Srinivas, A., Zito, P., Spencer, R. G. M., McFarland, J., Douglas, T. A., Conaway, C. H., Waldrop, M., & Mackelprang, R. (2020). Life at the Frozen Limit: Microbial Carbon Metabolism Across a Late Pleistocene Permafrost Chronosequence. *Frontiers in Microbiology*, *11*, 1753. https://doi.org/10.3389/fmicb.2020.01753

Natali, S. M., Watts, J. D., Rogers, B. M., Potter, S., Ludwig, S. M., Selbmann, A.-K., Sullivan, P. F., Abbott, B. W., Arndt, K. A., Birch, L., Björkman, M. P., Bloom, A. A., Celis, G., Christensen, T. R., Christiansen, C. T., Commane, R., Cooper, E. J., Crill, P., Czimczik, C., … Zona, D. (2019). Large loss of CO2 in winter observed across the northern permafrost region. *Nature Climate Change*, *9*(11), 852–857. https://doi.org/10.1038/s41558-019-0592-8

Richardson, D. C., Newbold, J. D., Aufdenkampe, A. K., Taylor, P. G., & Kaplan, L. A. (2013). Measuring heterotrophic respiration rates of suspended particulate organic carbon from stream ecosystems:

Measuring respiration rates of POC. *Limnology and Oceanography: Methods*, *11*(5), 247–261. https://doi.org/10.4319/lom.2013.11.247

Spencer, R. G. M., Mann, P. J., Dittmar, T., Eglinton, T. I., McIntyre, C., Holmes, R. M., Zimov, N., & Stubbins, A. (2015). Detecting the signature of permafrost thaw in Arctic rivers. *Geophysical Research Letters*, *42*(8), 2830–2835. https://doi.org/10.1002/2015GL063498

Shakil, S., Tank, S. E., Kokelj, S. V., Vonk, J. E., & Zolkos, S. (2020). Particulate dominance of organic carbon mobilization from thaw slumps on the Peel Plateau, NT: Quantification and implications for stream systems and permafrost carbon release. *Environmental Research Letters*, *15*(11), 114019. https://doi.org/10.1088/1748-9326/abac36

Tank, S. E., Vonk, J. E., Walvoord, M. A., McClelland, J. W., Laurion, I., & Abbott, B. W. (2020). Landscape matters: Predicting the biogeochemical effects of permafrost thaw on aquatic networks with a state factor approach. *Permafrost and Periglacial Processes*, *31*(3), 358–370. https://doi.org/10.1002/ppp.2057

Tanski, G., Wagner, D., Knoblauch, C., Fritz, M., Sachs, T., & Lantuit, H. (2019). Rapid CO2 Release From Eroding Permafrost in Seawater. *Geophysical Research Letters*, *46*(20), 11244–11252. https://doi.org/10.1029/2019GL084303

Turetsky, M. R., Abbott, B. W., Jones, M. C., Anthony, K. W., Olefeldt, D., Schuur, E. A. G., Grosse, G., Kuhry, P., Hugelius, G., Koven, C., Lawrence, D. M., Gibson, C., Sannel, A. B. K., & McGuire, A. D. (2020). Carbon release through abrupt permafrost thaw. *Nature Geoscience*, *13*(2), 138–143. https://doi.org/10.1038/s41561-019-0526-0

Zolkos, S., Tank, S. E., & Kokelj, S. V. (2018). Mineral Weathering and the Permafrost Carbon-Climate Feedback. *Geophysical Research Letters*, *45*(18), 9623–9632. https://doi.org/10.1029/2018GL078748

Zolkos, S., Tank, S. E., Striegl, R. G., & Kokelj, S. V. (2019). Thermokarst Effects on Carbon Dioxide and Methane Fluxes in Streams on the Peel Plateau (NWT, Canada). *Journal of Geophysical Research: Biogeosciences*, *124*(7), 1781–1798. https://doi.org/10.1029/2019JG005038

Vonk, J. E., Tank, S. E., Mann, P. J., Spencer, R. G. M., Treat, C. C., Striegl, R. G., Abbott, B. W., & Wickland, K. P. (2015). Biodegradability of dissolved organic carbon in permafrost soils and aquatic systems: A meta-analysis. *Biogeosciences*, *12*(23), 6915–6930. https://doi.org/10.5194/bg-12-6915-2015

---

## Author Comment (AC2)

**Reviewer Comment 2**

We would like to thank the reviewer for their careful comments, and also for this positive assessment of the manuscript.

**General Comments**

The study utilizes experimental aerobic incubations of sediments taken from slump affected streams in the Peel Plateau to investigate if the potential of mineralization of slump derived POC varies from that or POC in non-impacted streams, and to quantify the biodegradability of slump POC fractions relative to their transport potential.

Several experiments involving samples collected over different sites and seasons. In 2015 samples were collected from sediments in streams near and within different slump sites to test if slumps affected the biodegradability of POC. Water samples were also collected above, within and downstream of slumps. Water samples UU unfiltered upstream (in situ POC) relative to filtered upstream water to which slump POC was added (SU). In 2016 samples of sediment were collected near the SE slump, upstream, in the slump, and downstream of the slump – to test variability in biodegradability with transport.  In 2019 sediments were collected within and downstream of slump FM3 for follow-up experiments.

The authors conclude that there is minimal (4%) mineralization (oxidation) or POC over 1 month incubations. The authors propose that these low rates may be due in part to protection by adsorption to mineral particles. Additionally, the authors propose that the surrounding mineral rich tills promote inorganic C sequestration via chemolithoautotrophic processes.

This study involves the application of a carefully executed field sampling design, combined with carefully designed laboratory experiments and sophisticated analytical tools to address a very important knowledge gap in our understanding of the biogeochemical controls on the fate of particulate carbon released from permafrost thaw and disturbances. This is a study very worthy of publication. The methods are well detailed and documented, and overall the results are very well presented, although I have some concerns and suggestions about results. This is a very data/results rich paper.

My only substantive concern is the brevity of the conclusions. I feel the authors have missed the opportunity to really put these findings in to context. The authors show that these systems release and move a lot of carbon, but that the GHG emission potential is minimal – this is very significant, and should be discussed in the context of other work that suggests these abrupt thaw events might account for a large part of emissions from thawing permafrost - For example, the authors should discuss the meaning of their results in the context of the findings of Turetsky et al.'s Nature Geoscience, 2020 article (https://doi.org/10.1038/s41561-019-0526-0).

Thank you very much for these comments. We have now added a paragraph in our discussion section and modified our conclusion to place our findings within the larger context of permafrost

carbon release, including comparison to parameters used by Turetsky et al. (2020) highlighting that our highest estimate of 7% of TOC loss in this study is substantially lower than the estimate by Turetsky et al. (2020) that 2/3rds of DOC/POC from hillslope abrupt thaw will be mineralized. We compare our findings to the studies their estimate is based on and note that there are few studies that quantify rates of permafrost carbon mineralization within fluvial networks, and even fewer that include POC in these assessments. Therefore, current/early estimates of permafrost carbon feedbacks that include rates of fluvial C mineralization likely have a high degree of uncertainty and we recommend further studies, including POC and assessments of other carbon pathways, to create region-specific assessments of carbon release via hillslope abrupt thaw.

**Specific Comments**

This was a remarkably comprehensive and carefully designed and executed series of experiments. Although, I really appreciate the very concise explanations of the experiments in section 2.2, the subsequent results (sample acronyms and experiments) were hard to keep straight, until I read through the supplemental and saw Figures S5, S6, S7. I strongly recommend including Figures S5, S6, and S7 (or maybe some reduced form of one or two these) in the methods section of the main paper. These are great illustrations. I realize that space constraints might make this an editorial decision, however I found these figures were critical to clearly communicating the methods and experimental design. These will also help the reader keep the acronyms for the samples straight.

Thank you! We can move the 2015 and 2016 supplementary flow diagrams into the methods section of the main text. An additional option is to have them in the appendix area so that they can still be referenced within the main text body but not taking up space before the results. However, we don't think that moving the 2019 experiment flow diagram into the methods is necessary because the design is a replicate of 2015 SD treatment (as mentioned in the text), only with the addition of the sterilization treatment, so it is quite simple.

Table 1 is a really great help for summarizing the findings, and the limitations. However, it is a bit difficult to follow in places, I suggest presenting this table in landscape format, so that it is not as crowded. Again I realize this may be an editorial decision, however some rearrangement or reformatting is required to really maximize the readability of this important table.

We have altered Table 1 to landscape format, also in response to Reviewer 1's comment.

**Results**

In section 3.1 the authors report that the %change in POC is lower where the slump particles were added, and that this is likely due to the fact that the particle concentrations were so high in those samples that the % change is small. The % changes is potentially masking the importance of the magnitude the change in total mass of POC. It would be useful (more useful) to provide tables (or figures), in not in the main paper, then at least as a supplement that illustrates the changes in DOC, POC and TOC in terms of total g of C. Perhaps the point can be made at least in part by referring to the data as shown in Figure 3a and/or 3d for the DTOC.

Thanks for this comment. The percent changes are reported to assess the relative biodegradability. It is a common standardized metric used to assess biodegradability of organic carbon since larger initial organic carbon amounts would result in larger losses (Vonk et al. 2015). We do show absolute TOC changes relative to absolute oxygen changes in Figure 3 and we discuss it across all experiments and in reference to blank measurements in section 3.4.

Similarly, for the fractionated vs. unfractionated experiment. It would be helpful to show somewhere (e.g. Table B2) how the mass of C is distributed across the size fractions, to know where the greatest total C losses and gains are occurring, and thus to better interpret the % changes in terms of effect of size fractions on mass of C lost/gained. Perhaps this can be done in part by citing Figure 3b – which shows that the greatest change in C is due to the <20um fraction?

We have modified Table B2 to show that most of the carbon within the sample that was fractionated is associated with particles less than 20 µm (the smallest size fraction) and have added this statement into section 3.3.

"The majority of POC (73%, Table B2) was associated with particles less than 20 µm."

Section 3.4 does discuss the absolute carbon changes and we prefer it in this section because it compares the changes to oxygen changes.

The references to Appendices seemed odd to me. I am not familiar with Journals that support appendices, so I didn't even know where to look for them at first – I was happy to see they were at the end of the main document.. I think they definitely ought to appear in the main paper somehow – rather than in a supplemental -given that these data are very important in terms of the support they lend to the findings. The only exeption might be the material in Appendix C, which could go in the supplemental if necessary.

These appendices were created following the guidelines on the Biogeosciences website, which allows for both Appendices at the end of the manuscript, and a separate online-only Supplement (see https://www.biogeosciences.net/submission.html#manuscriptcomposition). We felt the content in the Appendices were more technical details that detracted from the main point of the paper that we were not finding high biolability of organic carbon mobilized from slumps to streams despite consistently rapid oxygen consumption rates across a suite of experiments, which was a surprise to us and not what the experiments were originally designed for (hence the follow-up in 2019). Much of the content in Appendix A (ANOVA tables) and Appendix B (characterization of size fractions which we believe to be ancillary) are reported in the main text. We would prefer to keep it there since our main text will lengthen with the addition of the flow diagrams.

**Technical Corrections**
Line 50: I suggest including some years to provide reader with more confined age of the Pleistocene age tills in this area, if known.

The tills were deposited as the Laurentide Ice Sheet advanced onto the eastern slopes of the Richardson mountains (Kokelj et al. 2017). We added in the period of maximum westward extent of the Laurentide Ice-sheet (c. 18 ka cal year BP (Lacelle et al. 2013)) and the point after which the Plateau likely was ice-free (13 ka cal year BP, Zazula 2009).

Line 52: Insert "the" after comma following "Thus, the relative.."

Done

Line 53:  Delete "Variations in", start sentence with "**S**ource composition can also vary…"

Done

Line 56 : Since there are 4 sites on the map in figure 1, I suggest inserting the site names of the 3 sites in brackets in this sentence to clarify the sites sampled for this experiment "In 2015, …within three slump sites **(HA, HB, HD**)

Done, also in response to Reviewer 1's comment

Figure 1: label all panels in the figure. E.g. the map should be labelled as panel a) then the headwall units panel b); and the sampling site locations panel c).

Done

Line 78: it is unclear how much water was used, the serum bottles were 120ml, but does this mean you used 120ml of water + 2 ml of slump runoff? Insert sample volume to be clear how much headspace was left in the bottles. e.g. "we incubated <**xx ml>** unfiltered upstream …."

The bottles were filled to have no headspace. This was highlighted in the supplementary but this detail is now added at the end of section 2.2.1 and 2.2.2.

Lines 85-87: The sieving process could use some additional explanation. It is not at all clear how such a small sample volume (0.5ml) could be sieved. Also were the fractions weighed? How did you know the mass of each fraction added (or the concentrations) of the final 60ml solution?

The fractionation process is detailed in supplementary S2 which has been slightly modified for clarity. The sediment concentrations (as highlighted in Figure 3e and Table S2) are abnormally high in this system. For the site where the sample was obtained (within slump SE) total suspended sediment concentrations can easily be upwards of 100 and 200 **grams** per litre. Thus, for these experiments (that were aiming to mimic downstream concentrations as shown in Table S2) we did not need to sieve more than 0.5 mL, which was still quite time consuming and difficult given the sediment load. We did not weigh fractions, we tried to approximate based on volume. The unfractionated treatment used slump runoff that was diluted to approximate downstream concentrations (Table S2) and so each fractionation was diluted to make the initial 0.5 mL used for fractionation up to a similar level of dilution. We tried to minimize loss of particles during the fractionation process but comparison of the sum of TSS across fractions to

the unfractionated shows there was error in the process (TSS sum across fractionations is 61% of TSS of unfractionated). However, we still think the relative proportions of material in different fractions is representative of the environment and this loss doesn't affect comparisons of % change which would compare the relative biodegradability of material in the different fractions (our initial aim of the experiment). Furthermore, the unfractionated control is presented alongside all results.

Line 106: Delete "First," and start this sentence with "To assess…"

Done

Line108: Insert the volume of sample used, so that water vs. headspace volume in the 60ml bottles is clear. "We incubated **"XX mL"** of sample in 60mL glass BOD bottles

We addressed this comment above by noting all bottles were filled to not have headspace. Comparison of concentrations to *in situ* concentrations are shown in Table S2.

Line 113: The bracket should include reference to equations 1a, 1b. "…(eqns. **1**-4)…" x

References have now been added to each equation bracket, with references used below (for full refs see reference list at the end).

Equations 1a and b (Stumm and Morgan 2012)

Equation 2 (Klatt and Polerecky 2015)

Equation 3 (Percak-Dennett et al. 2017)

Equation 4 (Zolkos and Tank 2020, Calmels et al. 2007)

Line 114: Indicate the methods used to quantify N species and sulfate, and/or refer to citation or supplemental where this is explained at end of sentence.

This detail has now been added with reference to supplementary 3.3 which has been modified to provide more detail on the analysis.

Line 113: Since your goal as stated at the start of section 2.2.4 is to asses O2 losses and OC gains, and since not all the equations (1-4) contribute to generating OC. You should insert "could consume $O_2$" in this sentence.  E.g. "…could **consume $O_2$** and/or generate OC, …".

We have modified the sentence.

**Results**
Line 141. I am not familiar with having appendices in journal articles. I suggest adding material from Appendix A to the main paper or the supplemental.

Please see our comment regarding this above

Line 145: It would be helpful to show the DSUVA254, in the paper or in the supplemental.

The change in SUVA is provided in Appendix C (Table C1).

line 160: Figure 2 caption, you say "measured (point) and modelled (line) O2" but there are no points visible in panels a-c.

We have amended the figure caption to show only modelled values are shown in Figure 2 (ease of readability in the Figure). Comparison of measure to modelled are shown in the supplementary (Figures S1 – S3).

Lines 185-191: Minor point, but you use lower case letters to identify the panels in Figure 3, yet in the text you cite Figure 3A, 3C etc. using upper case. I suggest that you should use lower case letters in the in text citations (Figure 3a, 3c…) to be consistent with the figures.

Done. If we missed any, we will re-address this during editorial reviews.

Line 202: Figure 3 caption, note the 1:1 line in panel (f) is solid vs. other panels where it is dashed. Is there a reason why this one is different? If so explain this in the caption, if not edit so that it is dashed as in the other panels.

Done

Line 215 and 218 – I suggest replacing "balancing to" with "resulting in"

We prefer "balancing to" because it is a balance

Line 218: Insert "in sterilized bottles" after TIC.

Done.

Line 223 and 227 – use lower case letters in reference to figure panels, so that these are consistent with how they appear in the figure.

Addressed above.

Line 228-230: I think this sentence requires rewording to clarify the message the authors are making. The authors suggest that the increase in simple compounds in sterilized samples "cautions against assuming the sterilized treatment is a true abiotic control of organic matter changes". This seems to suggest that you are calling into question the fact that your sterile samples were truly sterile, which I don't think is the intent. I think you mean to indicate that the changes in DOM could be entirely due to the sterilization process itself, hence the change in DOM composition of the sterile samples cannot be considered a "control" or "baseline" of the of DOM if there had been no biological activity in the samples. I suggest maybe a simple correction, delete

"PC1 separated DOM…proportion of simple compounds," **Given that the sterilization process itself could increase the proportion of simple compounds**, the results caution against …. ".

We edited the sentence to read:

"PC1 separated DOM in sterilized and unsterilized bottles, suggesting the sterilization processes increased the proportion of simple compounds. Since the sterilization process itself appears to increase the proportion of simple compounds, the results caution against its use as an abiotic baseline."

**Discussion**

Line 261-262: … indicate that $CO_2$ production ceased by the end of …" this sentence requires a citation to support this statement.

The reference is Tanski et al. (2019) and has been added (see reference list below).

Line 282: you suggest that chemolithoautotrophy as a possible mechanism for counterbalancing OC mineralization. Can you discuss or provide evidence to support that these reactions are likely/possible in these environments - i.e. are the thermodynamics/redox conditions consistent with environments where these chemolithoautotrophic processes (organisms) are known to occur?

Nitrification is a common process in freshwaters (Stumm and Morgan 2012) but we do discuss the slow growth of nitrifying microbes and how the stoichiometry doesn't balance to the gains we see in experiments (see paragraph 3 of the discussion). In mine tailings pyrite oxidation has been found to be a notable carbon sequestration process (Li et al. 2019). The process is commonly associated with acidic-pH conditions, but it can also occur at circum-neutral pH (e.g. Percak-Dennett et al. 2015). However, sulfide oxidation and associated carbon sequestration is a very diverse and complex process (Klatt and Polerecky, 2015) so the stoichiometry can range quite a bit, but given the broad prevalence of sulfides across the Peel Plateau we think it's a mechanism worth exploring. A paragraph in the discussion (with some additional edits added) is dedicated to this.

**Conclusions**
These findings of the low biolability of the permafrost POC are so important, yet the significance is not raised at all in the conclusions. These findings need to be put into context in the conclusions to better highlight their significance – as stated above especially with respect to Turetsky et al 2020. It seems to be broadly accepted that these large abrupt permafrost thaw events are likely to have strong positive feedbacks on atmospheric C and climate. Your study calls this into question – I think you need to highlight this.

We have addressed this comment and modified the conclusion as noted above.

**Supplemental Information**
**Page 3, Para 2: line 4:** "Material **the** passes through " replace "the" with "that"

Done

**Page 3, Para 2: line 6:** "…. Material passed through the filter **discard** and…"should be "discarded"

Corrected to, "Material that passed through the filter was discarded…"

**Page 3, Para 2: line 9:** INSERT "from the 0.5 mL sample" between "particles" and "through the …"

Done

**Section 3.3 paragraph 1 line 2**: a 0.7 micron filter is not standard for these analyses. Can you comment on why you used this pore size, and also what if any effect the larger pore size might have relative to standard measures?

This pore size is larger than the 0.45 micron pore size typically used for analysis of nutrients and trace metals. We filtered for PIC using a 0.7 GF/F filter (as required for carbon analyses) and then used the filtrate for dissolved analyses because we had a very limited amount of sample water available. A main point of concern with using a GF/F filter for chemical analysis, particularly for dissolved nutrients, might be greater bacterial activity since 0.7 microns may allow more microbes through than 0.45. This potential effect is likely limited because samples were immediately frozen and submitted for analyses within 2-4 days of collection. Further, we do not think the matrix would have changed substantially between filter sizes because colloidal organic matter (i.e. not truly dissolved material) still exists below 0.45 microns.

**Figure S5** – since you didn't use both time points, I suggest removing the one you didn't use. Also indicate the time of the timepoint (30 days?) on this and other figures or in the captions.

Data from timepoint 2 is available in the supplementary so we wanted to leave it in the figure for any interested readers. We now indicate in the caption that we focused our analyses on timepoint 1 for brevity, with results from timepoint 2 available in the supplement.

**References**

Calmels, D., Gaillardet, J., Brenot, A., & France-Lanord, C. (2007). Sustained sulfide oxidation by physical erosion processes in the Mackenzie River basin: Climatic perspectives. *Geology*, *35*(11), 1003. https://doi.org/10.1130/G24132A.1

Klatt, J. M., & Polerecky, L. (2015). Assessment of the stoichiometry and efficiency of CO2 fixation coupled to reduced sulfur oxidation. *Frontiers in Microbiology*, *6*. https://doi.org/10.3389/fmicb.2015.00484

Kokelj, S. V., Tunnicliffe, J. F., & Lacelle, D. (2017). The Peel Plateau of Northwestern Canada: An Ice-Rich Hummocky Moraine Landscape in Transition. In O. Slaymaker (Ed.), *Landscapes and*

*Landforms of Western Canada* (pp. 109–122). Springer International Publishing. https://doi.org/10.1007/978-3-319-44595-3_7

Lacelle, D., Lauriol, B., Zazula, G., Ghaleb, B., Utting, N., & Clark, I. D. (2013). Timing of advance and basal condition of the Laurentide Ice Sheet during the last glacial maximum in the Richardson Mountains, NWT. *Quaternary Research*, *80*(2), 274–283. https://doi.org/10.1016/j.yqres.2013.06.001

Li, Y., Wu, Z., Dong, X., Xu, Z., Zhang, Q., Su, H., Jia, Z., & Sun, Q. (2019). Pyrite oxidization accelerates bacterial carbon sequestration in copper mine tailings. *Biogeosciences*, *16*(2), 573–583. https://doi.org/10.5194/bg-16-573-2019

Percak-Dennett, E., He, S., Converse, B., Konishi, H., Xu, H., Corcoran, A., Noguera, D., Chan, C., Bhattacharyya, A., Borch, T., Boyd, E., & Roden, E. E. (2017). Microbial acceleration of aerobic pyrite oxidation at circumneutral pH. *Geobiology*, *15*(5), 690–703. https://doi.org/10.1111/gbi.12241

Stumm, W., & Morgan, J. J. (2012). *Aquatic Chemistry: Chemical Equilibria and Rates in Natural Waters*. John Wiley & Sons.

Tanski, G., Wagner, D., Knoblauch, C., Fritz, M., Sachs, T., & Lantuit, H. (2019). Rapid CO2 Release From Eroding Permafrost in Seawater. *Geophysical Research Letters*, *46*(20), 11244–11252. https://doi.org/10.1029/2019GL084303

Zazula, G. D., MacKay, G., Andrews, T. D., Shapiro, B., Letts, B., & Brock, F. (2009). A late Pleistocene steppe bison (Bison priscus) partial carcass from Tsiigehtchic, Northwest Territories, Canada. *Quaternary Science Reviews*, *28*(25), 2734–2742. https://doi.org/10.1016/j.quascirev.2009.06.012

Zolkos S and Tank S E 2020 Experimental Evidence That Permafrost Thaw History and Mineral Composition Shape Abiotic Carbon Cycling in Thermokarst-Affected Stream Networks *Front. Earth Sci.* **8** Online: https://www.frontiersin.org/articles/10.3389/feart.2020.00152/full